

# The role of ascent timescale for WCB moisture transport into the UTLS

Cornelis Schwenk [1] and Annette Miltenberger [1]

[1]Institute for Atmospheric Physics, Johannes Gutenberg University Mainz

**Correspondence:** Cornelis Schwenk (c.schwenk@uni-mainz.de)

**Abstract.** Warm conveyor belts (WCBs) are coherent ascending airstreams in extratropical cyclones. They are a major source of moisture for the extratropical upper troposphere and lower stratosphere (UTLS), where moisture acts as a potent greenhouse gas and WCB-associated cirrus contribute to cloud radiative forcing. However, the processes controlling WCB moisture transport and cloud properties are poorly characterised. Furthermore, recent studies have revealed (embedded) convection as a ubiquitous feature of WCBs, highlighting the importance of understanding their updraft and microphysical structure. We present a Lagrangian investigation of WCB moisture transport for a case from the WISE (Wave-driven ISentropic Exchange) campaign based on a convection-permitting simulation. Lagrangian non-dimensional metrics of the moisture budget suggest that the ascent timescale ($\tau_{600}$) strongly controls the end-of-ascent total moisture content, which is largest for slowly ascending trajectories ($\tau_{600} \geq 20\,\mathrm{h}$, 30 % of all WCB trajectories). This is due to relatively warm end-of-ascent temperatures and the strong temperature control on transported water vapor. Deviations from equilibrium water vapor - condensate partitioning are largest for slow trajectories due to faster glaciation and lower ice crystal numbers. A local moisture transport minimum at intermediate $\tau_{600}$ results from a shift towards a riming dominated precipitation formation pathway and decreasing outflow temperatures with decreasing $\tau_{600}$. The fastest trajectories ($\tau_{600} \leq 5\,\mathrm{h}$, 5 % of all WCB trajectories) transport the largest condensate mass to the UTLS due to less efficient condensate loss, and produce the longest-lived outflow cirrus. Models that parameterise convection may under-represent these processes, potentially impacting weather forecasts and climate predictions.

## 1 Introduction

Warm conveyor belts (WCBs) are regions of large-scale, coherent, ascending airflow in the vicinity of extratropical cyclones (ETCs), that produce the elongated cloud band typically associated with ETCs. WCBs usually transport moist boundary layer air pole-ward and into the upper troposphere over a span of two days, during which a host of complex physical processes take place that make WCBs important across planetary scales for Earth's weather, climate and radiative budget (Madonna et al., 2014).

As a common weather phenomenon, WCBs occur frequently and are major drivers of mid-latitude weather, influencing large-scale weather conditions for weeks after their dissipation (Joos et al., 2023; Madonna et al., 2014; Rodwell et al., 2018). The ascending air masses undergo diabatic heating, which can produce potential vorticity (PV) anomalies. These anomalies affect





cyclone strength and lifetime (Binder et al., 2016; Rossa et al., 2000), modify the upper-level wave guide and jet stream (Pickl et al., 2023; Grams et al., 2011; Joos and Wernli, 2011; Wernli, 1997), and have been linked to atmospheric blocking (Pfahl et al., 2015; Wandel et al., 2024). Additionally, WCBs contribute to over 80% of the total precipitation in Northern Hemisphere storm tracks (Pfahl et al., 2014; Eckhardt et al., 2004). WCBs are therefore crucial for predicting extreme weather events such
as heat waves and storms (Oertel et al., 2022; Flaounas et al., 2017), and the incorrect representation of WCBs has been identified as a key factor in amplifying forecast uncertainties (Pickl et al., 2023; Berman and Torn, 2019).

As a large-scale climatological phenomenon, WCBs significantly influence Earth's climate. As the warm, planetary boundary layer (PBL) air ascends into the upper troposphere/lower stratosphere (UTLS), a multitude of warm-phase, mixed-phase,
and cold-phase microphysical processes occur, producing various cloud types. Each of these cloud types, with distinct microphysical properties, affects Earth's radiative budget through cloud radiative forcing (CRF) (Joos, 2019). Determining the net contribution to CRF is complex and has been investigated in several studies (Joos, 2019; Gehring et al., 2020; Stewart et al., 1998). The long-lived cirrus shield that accompanies a WCB in the later stages of its life cycle is optically thin and has low cloud-top temperatures, resulting in positive CRF (Spichtinger et al., 2005; Binder et al., 2020; Joos, 2019). Conversely, the
warm-phase, low-altitude clouds that form in the early stages of a WCB's life cycle at lower latitudes are optically thick and have high cloud-top temperatures, resulting in negative CRF (Joos, 2019). Mixed-phase clouds, with varying optical thickness and temperature, have an uncertain CRF sign. Therefore, the net CRF of WCBs and their impact on Earth's radiative budget are sensitive to the microphysical representation of clouds produced in WCBs (Joos, 2019).

In addition to CRF, WCBs influence Earth's radiative budget because they transport water vapour into the UTLS. This happens even though the moisture content of ascending air in a WCB is greatly reduced by the formation of precipitation (Madonna et al., 2014; Sprenger and Wernli, 2003). Water vapor is the most dominant greenhouse gas, and studies have shown that UTLS water vapor, in particular, is the most significant positive feedback factor in climate change (Li et al., 2024; Held and Soden, 2000). Even minor changes in UTLS specific humidity can have a substantial impact on the greenhouse effect (Wang et al.,
2001; Hansen et al., 1984). WCBs have generally been associated with significant troposphere-stratosphere exchange (Gettelman et al., 2011), but so far, no studies have quantified the amount of vapour transported into the UTLS by WCBs. Only Zahn et al., 2014, who investigated the origins of measured UTLS water vapour using backwards trajectories, determined that WCBs are one of four dominant transport pathways for water into the UTLS. It also remains unclear how much moisture during the WCB ascent is converted into precipitation, which microphysical processes are responsible for this conversion, and to what
extent each process contributes.

Given the importance of WCBs, it is essential that weather and climate models accurately represent them. However, it remains unclear what uncertainties are introduced by the parameterisation of microphysical processes when modelling WCBs. These parameterisations are known to introduce significant uncertainties when modeling storms and convective systems (Barthlott
et al., 2022; Khain et al., 2015; Dearden et al., 2016; Hieronymus et al., 2022) and sensitivity experiments have shown that





the choice of microphysical parameterization scheme impacts the evolution of WCBs (Mazoyer et al., 2021, 2023). However, the uncertainties introduced by these parameterisations regarding WCB vapor transport into the UTLS and cloud formation, for example, have not been quantified. Consequently, the uncertainty of the CRF induced by WCBs remains largely unknown. Oertel et al., 2023 investigated which microphysical processes produce diabatic heating in a WCB and how these differ for quickly and slowly ascending WCB air parcels. However, it has not yet been studied which microphysical processes modify vapor and hydrometeor content in ascending WCB air, meaning that the uncertainties introduced for the transport of vapour are also unknown.

Another crucial aspect of WCBs that remains poorly understood is the role of deep and embedded convection for the transport of vapour and the production of clouds. Traditionally, WCBs are viewed as consisting of a coherent, slowly and slantwise ascending air stream. However, recent studies using observations and high-resolution simulations suggest that convection is a common phenomenon in WCBs (Binder et al., 2016; Oertel et al., 2019, 2020, 2021; Rasp et al., 2016) and that WCBs potentially produce extreme precipitation in regions of deep convection (Flaounas et al., 2017). This is significant because deep convection could be a major source of stratospheric ice clouds (Zou et al., 2021) and has been found to transport more tracers into the upper atmosphere than large-scale advection (Purvis et al., 2003). Forecast errors have also been found to grow more quickly in regions with convective activity (Rasp et al., 2016). It has not been studied how convective regions in a WCB transport water vapor and hydrometeors into the UTLS differently than regions of slantwise ascent, nor has the extent of convection's contribution to this transport been quantified. What has been shown is that higher-resolution models produce Lagrangian WCB air parcels that ascend faster, reach higher altitudes, and experience more latent heating (Choudhary and Voigt, 2022). This presents a challenge for the most commonly used models that parameterise convection rather than explicitly treating it, as they may misrepresent or under-represent processes associated with convection. Such models might incorrectly model the formation of clouds and the transport of moisture into the UTLS, which would impact the accuracy of weather forecasts and predictions of future UTLS water vapour content.

This paper aims to address these knowledge gaps by investigating the microphysical processes that control WCB moisture transport into the UTLS and quantifying how these processes differ between regions of convective and non-convective activity. To this end, we conduct a case study of a North Atlantic WCB using a convection permitting ICON simulation (using the two moment microphysics scheme from Seifert and Beheng, 2005) and investigate both Eulerian and Lagrangian output fields.

The paper is structured as follows. First, we describe our ICON model setup and the online trajectory module it utilises. Next, we present our novel Lagrangian methods for investigating microphysical processes during WCB ascent of air parcels. This section introduces and derives Lagrangian formulations for common diagnostics, such as drying ratio and precipitation efficiency. We then present our case study and describe the WCB in question. In our results, we first examine ascent diagnostics to quantify the prevalence of convection in the WCB, and then focus on moisture transport. We analyse several diagnostics at (i) the beginning of the ascent, (ii) the end and hours after the ascent, and (iii) during the ascent, to provide a comprehensive





picture of the moisture transported by the WCB. Finally, we summarise our findings and conclude with the implications of this study for future research on WCBs.

## 2 Methods

In this chapter, we first describe the ICON model setup used for our simulations. We then outline our WCB trajectory selection algorithm and explain our method of normalising the ascent time axis around the WCB ascent of a trajectory. Finally, we give a detailed overview of the variables used in our investigations. In particular, we derive Lagrangian formulations for the drying ratio, precipitation efficiency, and condensation ratio, among others.

### 2.1 ICON model setup

Our case study considers a WCB that dominated weather over the northern Atlantic ocean on 23 September 2017. To simulate the evolution of the atmosphere for this case study we used the ICOsahedral Nonhydrostatic (ICON) modelling framework (version 2.6.2; Zängl et al., 2014). Our simulation was initialised with the operational ICON global analysis at 00:00 UTC 20 September 2017, a couple of hours before the cyclone formed. The simulation ran for 96 h until 00:00 UTS 24 September 2017, at which time the WCB dissipated over northern Europe.

In addition to a computation domain that spans the entire globe, ICON can embed regions of higher spatial resolution (called "nests") within the global grid. ICON includes an implementation of two-way nesting, meaning that simulations on lower resolution domains couple to the next highest resolution simulation by providing lateral boundary conditions, and are in turn nudged towards the solutions of the higher resolution Zängl et al., 2022. Our model ran with two nested domains. Computations on the global domain were conducted with a 120 s time step on a R03B07 grid (effective grid spacing of approximately 13 km), and on the nested domains with 60 s and 30 s time steps on on R03B08 (∼6.5 km) and R03B09 (∼3.3 km) grids, respectively. The nests were chosen such that they cover the main region of WCB ascent, which was visually determined before the model run using cloud cover and sea-level pressure data from ERA5. Figure 1 shows the nested setup with a snapshot of ERA5 reanalysis data (Hersbach et al., 2020) over the main WCB ascent region for 10:00 UTC 23 September 2017. We can see that the main part of the WCB falls into the nested domains.

We chose the spatial resolutions of the highest-resolution nested domain such that it permits convection. On the global domain, convection is parameterised using the Tiedtke-Bechtold convection scheme (Tiedtke, 1989; Bechtold et al., 2008), whereas on the nested domains only shallow convection is parameterised. On all domains non-orographic gravity wave drag, sub-grid scale orographic drag (Lott and Miller, 1997; Orr et al., 2010) and turbulence are parameterised using the standard ICON schemes. Radiation is treated using the Rapid Radiative Transfer Model. Cloud microphysical processes, which are central to this paper, are calculated using the two-moment microphysics scheme by Seifert and Beheng (2005). This scheme represents hydrometeor mass mixing ratios $q_x$ and the corresponding number concentrations $n_x$ for six hydrometeor species: cloud droplets (c),



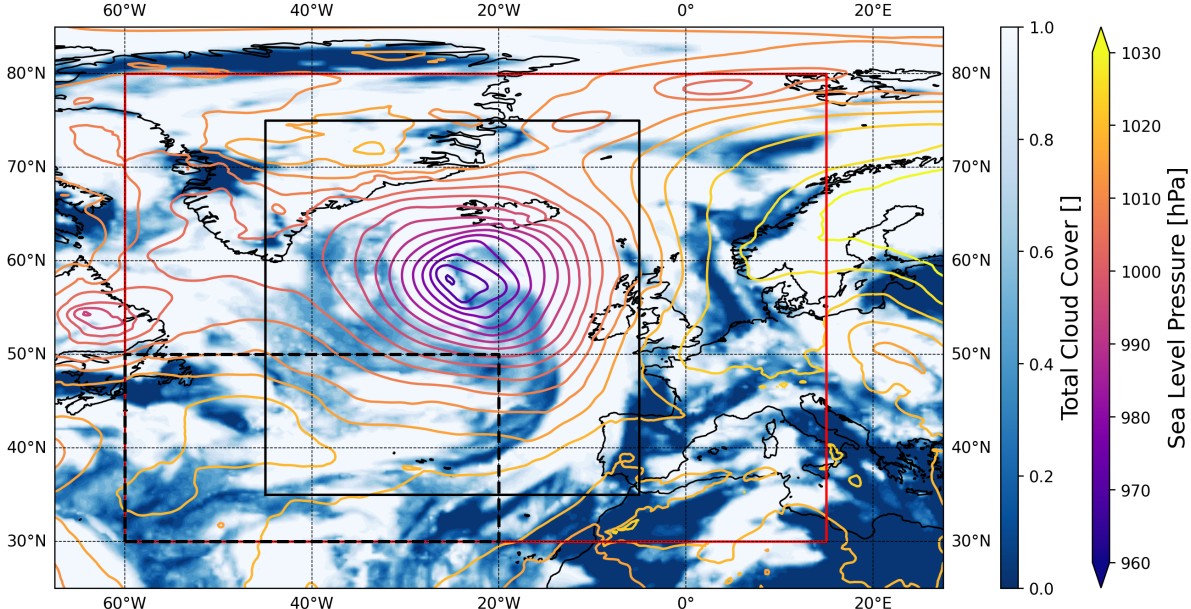

**Figure 1.** Snapshot of the atmosphere at 10:00 UTC 23 September 2017 using ERA5 data. The nested domain boundaries are shown for the first nest (red) and the second nest (black). The starting area of the online trajectories is also outlined (black dashed). Through the sea-level pressure contours and the total cloud cover one can clearly see the prominent WCB cloud band and low pressure system.

rain drops (r), ice crystals (i), snow flakes (s), graupel grains (g), and hailstones (h). The model setup we have described here allows a detailed representation of the region of interest, i.e. the WCB ascent and initial outflow, while offering a consistent representation of the larger-scale surroundings of the WCB.

### 2.1.1 Online trajectories

Lagrangian data can be computed online (during a model simulation, using wind fields computed at each time step) or offline (after a model simulation, using Eulerian output files). Online trajectories offer accurate solutions to the trajectory equation for high-resolution models and detailed perspective at high temporal resolution on e.g. microphysical process rates (Oertel et al., 2023; Miltenberger et al., 2020, 2016, 2013). We therefore used an online trajectory module implemented in ICON by Miltenberger et al., 2020 and extended by Oertel et al., 2023 to support domain-nesting. We started online trajectories every 5 h during the 96 h simulation and data was written to file every 30 min. Starting positions were chosen at random from within the starting region (shown using the black dashed lines in Figure 1) and from six vertical model levels (ranging between ca. 1000 and 800 hPa). We chose the starting region, which spans across the two nested domains, on the basis of prior offline trajectories that were calculated using ERA5 reanalysis data. This ensured that we obtained plenty of trajectories representing the WCB airstream.





## 2.2 Warm conveyor belt selection

Our selection algorithm identified WCB trajectories conditional to them fulfilling two criteria. Firstly, a WCB trajectory must ascend at least 600 hPa in at most 48 h. This is a widely used criterion/limit imposed by previous studies (e.g., Madonna et al., 2014; Oertel et al., 2023; Rasp et al., 2016). Secondly, all WCB trajectories must be within two visually determined lon-lat regions at two certain times. These regions were determined on the basis of Eulerian cloud cover and sea level pressure data from our simulation. This ensured that the trajectories were not part of a mesoscale convective system (MCS) unrelated to our WCB or any other cyclone.

## 2.3 Ascent timescales and characterisation of convective behaviour

We are interested in the ascent timescales of trajectories because we want to investigate the role of embedded convection and vertical velocity on the microphysical processes governing moisture transport. Different vertical velocities impact the dominant precipitation formation pathway and thereby likely the efficiency of precipitation formation. In line with this expectation, Oertel et al., 2020 found varying hydrometeor compositions for trajectories with differing $\tau_{600}$ indicating that different microphysical processes dominant the microphysical evolution during ascent. In addition, Oertel et al., 2023 also found that the microphysical processes contributing to adiabatic heating are different for different ascent timescales. They however did not quantify what these differences mean for the moisture budget. In the following we introduce the ascent time scales and diagnostics we use to characterise convective behaviour of WCB trajectories.

The simplest and most widely used ascent timescale is the fastest time in which a trajectory ascends 600 hPa, called $\tau_{600}$ (Rasp et al., 2016; Oertel et al., 2023). We use this timescale to differentiate convective (small $\tau_{600}$) from slantwise ascending (large $\tau_{600}$) WCB trajectories. We are however interested in the entire WCB ascent, not just the section where the ascent is fastest. We therefore also look at the time for which the ascent velocity remains above 8 hPa/h before and after the $\tau_{600}$ time segment, and call this time $\tau_{WCB}$. We use $\tau_{WCB}$ to define the period during which a trajectory is ascending as part of the WCB; before/after this time it is defined to be in the inflow/outflow.

To get a broader picture of the overall convective activity, we also consider additional variables. Similarly to $\tau_{600}$, we also look at the 300 hPa and 400 hPa ascent times ($\tau_{300}$, $\tau_{400}$), to quantify the convective behaviour for less deep ascent (i.e embedded mid-level convection). We also look at the maximum 2-hour pressure difference ($\max(\Delta p_{2h})$) and the minimum 10-hour pressure difference ($\min(\Delta p_{10h})$) during the WCB ascent, to assess the "regularity" with which trajectories rise. The last metric of course only makes sense for trajectories that take more than 10 h to ascent.

The metrics introduced here allow for a deeper insight into a trajectory's ascent behaviour. Consider for instance the three most important metrics: $\tau_{600}$, $\tau_{WCB}$ and $\max(\Delta p_{2h})$, visualised by a cartoon in Figure 2. In this example, a WCB trajectory is ascending from the boundary layer at around 950 hPa to 350 hPa over a time of approximately 28 h. The ascent velocity re-





mains above 8 hPa/h ($\tau_{\mathrm{WCB}}$) for 14 h and is colored yellow. The minimum time taken for a 600 hPa ascent along the trajectories entire path ($\tau_{600}$) is 8 h and is colored red. The maximum pressure difference experienced by the trajectory in any 2 h time span $(\max(\Delta p_{2\,\mathrm{h}}))$ is 350 hPa and is colored cyan. This trajectory would be described as ascending in a purely slantwise fashion if only $\tau_{\mathrm{WCB}}$ is considered, but $\max(\Delta p_{2\,\mathrm{h}})$ shows that there is a short period of convective ascent. It is therefore important to consider multiple metrics to correctly classify and understand the ascent behaviour of a trajectory.

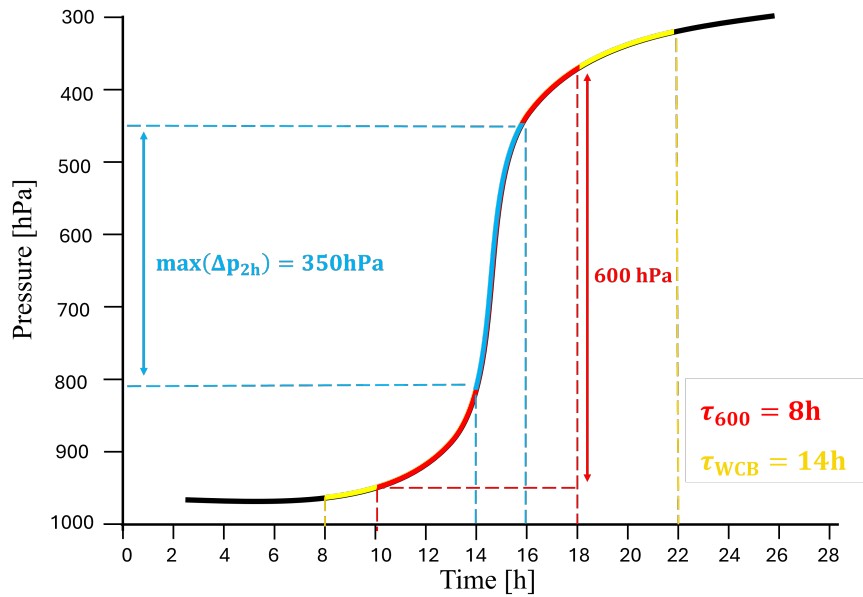

**Figure 2.** Visualisation of the three most important ascent metrics, $\tau_{600}$ (red), $\tau_{\mathrm{WCB}}$ (yellow) and $\max(\Delta p_{2\,\mathrm{h}})$ (cyan), that help to give a full picture of a trajectory's ascent behaviour.

### 2.4 Normalised ascent timescale

We want to be able to compare the physical processes of WCB trajectories with different ascent timescales at similar stages of their ascent. This becomes difficult when looking at data along a $\tau_{\mathrm{WCB}}$ time axis, as fast trajectories will complete their ascent and reach the UTLS at a time when slow trajectories may still be in the lower troposphere. We therefore normalise the $\tau_{\mathrm{WCB}}$

ascent time for all trajectories and introduce the normalised ascent time $\tilde{t}$. The time at which trajectories begin to ascend ($t_0$) is set to zero ($\tilde{t} = 0$), and the time at which they complete the ascent ($t_{\mathrm{end}} = t_0 + \tau_{\mathrm{WCB}}$) is set to one ($\tilde{t} = 1$). Now all normalised ascent times are bound by 0 and 1 and any two trajectories at a given normalised time are likely to be at similar stages of their ascent. This approach allows for an investigation and comparison of physical processes along a coherent and ascent-oriented time axis.





## 2.5  Time-integrated rates

We consider time-integrated (as opposed to instantaneous) microphysical process rates. This is done so that we can determine the accumulated effects of a microphysical process along a trajectory during its WCB ascent. ICON computes instantaneous microphysical process rates $\psi_{\mathrm{inst}}(t)$ with units $[\psi_{\mathrm{inst}}(t)] = \mathrm{kg\,kg}^{-1}\mathrm{s}^{-1}$ (see supporting information). We use these to define the time-integrated rates that are calculated by the online trajectory module using the spatially interpolated instantaneous rates at each model physics timestep:

$$\psi(t) = \int_0^t \psi_{\mathrm{inst}}(t')dt', \qquad [\psi] = \mathrm{kg\,kg}^{-1}. \tag{1}$$

To focus on the WCB ascent only, i.e. ignoring anything that happened before, the time-integrated rates are set to zero (in post-processing) at the start of the WCB ascent, i.e. $\psi(\tilde{t} = 0) = 0$.

## 2.6  Moisture budget

To understand the physical mechanisms that control the final moisture content in the outflow of a WCB, we will now define variables that describe the specifics of removing (or adding) moisture from (to) an air parcel.

### 2.6.1  Lagrangian Drying Ratio

In the most general sense, the efficiency with which moisture is removed from a parcel during a WCB ascent can be described by the drying ratio (DR) (e.g., Miltenberger, 2014). It quantifies how much moisture initially contained in the Lagrangian parcel is *removed* by the end of the ascent and is given by:

$$\mathrm{DR} = \frac{Q_{\mathrm{tot}}(0) - Q_{\mathrm{tot}}(1)}{Q_{\mathrm{tot}}(0)} \qquad \Big(\text{note}: \quad Q_{\mathrm{tot}}(1) = Q_{\mathrm{tot}}(0) \cdot (1 - \mathrm{DR})\Big). \tag{2}$$

Here $Q_{\mathrm{tot}}(\tilde{t})$ is the total moisture content at the beginning ($\tilde{t} = 0$) and at the end ($\tilde{t} = 1$) of the WCB ascent so the numerator is equal to the change in total moisture during the ascent. $Q_{\mathrm{tot}}(\tilde{t})$ is the sum of the specific humidity (qv) and all hydrometeor mass mixing ratios (qc, qr, qi, qs, qg and qh, see Appendix Section B). This definition of DR is a Lagrangian one, because we consider the differences in moisture along the trajectories.

DR is non-zero if moisture is removed from a trajectory by the end of the ascent (DR=1 if all moisture is removed). The mechanisms for moisture removal in our simulation are (i) the turbulence parameterisation, (ii) the convection parameterisation, (iii) precipitation and (iv) numerical uncertainties/interpolation errors. We can therefore rewrite Equation 2 as follows:

$$\mathrm{DR} = \frac{Q_{\mathrm{tcr}}(1) + P(1)}{Q_{\mathrm{tot}}(0)}. \tag{3}$$

In this equation, $Q_{\mathrm{tcr}}(1)$ is the sum of the moisture removed by the turbulence and convection parameterisations and the numerical uncertainties by the end of the ascent (for details see Appendix Section B). We refer to this as the moisture loss due



the "mixing" processes. $P(1)$ is the time-integrated net precipitation rate at the end of the ascent, which accounts for the mois-
ture lost by the gravitational removal of hydrometeors from the parcel (see Appendix Section B). Note that this is not surface
precipitation, but the net flux of hydrometeors leaving the parcel.

We can conclude that the numerator terms in Equation 3 include all the mechanisms in our simulation that can remove mois-
ture from a trajectory. Their sum is therefore equal to the change in total moisture during the ascent and equation 3 is equal to
equation 2.

### 2.6.2   Lagrangian microphysical and mixing drying ratio

More detailed insight into the processes that determine DR can be gained by splitting DR into terms that describe different
moisture removal mechanisms. First we define the mixing drying ratio:

$$\mathrm{DR_{mix}} = \frac{Q_{\mathrm{tcr}}(1)}{Q_{\mathrm{tot}}(0)}, \tag{4}$$

which describes what fraction of initial moisture is lost due to mixing processes. All remaining moisture $(Q_{\mathrm{tot}}(0) - Q_{\mathrm{tcr}}(1))$
either remains in the parcel or is removed by precipitation, which brings us to the definition of the microphysical drying ratio:

$$\mathrm{DR_{mphys}} = \frac{P(1)}{Q_{\mathrm{tot}}(0) - Q_{\mathrm{tcr}}(1)}. \tag{5}$$

$\mathrm{DR_{mphys}}$ is the fraction of this remaining moisture that is removed by the precipitation of hydrometeors. Its name comes from
the fact that moisture loss due to precipitation depends on the microphysical processes that form and grow hydrometeors.


For completeness, note that the denominators of the two drying ratios we have defined are not the same. Our choice of denom-
inators ensures that $\mathrm{DR_{mphys}}$ only describes moisture removal by precipitation *after* the removal by mixing processes is taken
into account. $\mathrm{DR_{mphys}}$ is not 1 if all the moisture has been removed by precipitation alone, but if all the moisture that has
not been removed by turbulence/convection has been removed by precipitation. This choice reflects the focus of this paper on
microphysical processes. Regardless of the choice of denominator, DR can be recovered from $\mathrm{DR_{mix}}$ and $\mathrm{DR_{mphys}}$ as follows:

$$\mathrm{DR} = \mathrm{DR_{mix}} + \mathrm{DR_{mphys}} - \Theta, \quad \text{with} \quad \Theta := \mathrm{DR_{mix}} \cdot \mathrm{DR_{mphys}}. \tag{6}$$

The interaction term $\Theta$ results from the fact that both moisture removal by mixing and precipitation occur simultaneously, i.e.
any moisture removed by turbulence/convection is not available for precipitation. We now shift our focus to the processes that
govern $\mathrm{DR_{mphys}}$.

### 2.6.3   Lagrangian Precipitation Efficiency (PE) and Condensation Ratio (CR)

If water vapour is to be removed from an air parcel by precipitation, it must first be converted into cloud condensate. This
can occur either by the formation of new hydrometeors (nucleation) or by the growth of hydrometeors from the gas phase



(deposition and condensation) that were either already present in the parcel, sedimented into the parcel, or were mixed by
turbulent processes into the parcel. This first step can be quantified by the Condensation Ratio (CR) (e.g., Barstad et al., 2007;
Miltenberger, 2014). In the second step, condensate may leave the parcel as precipitation. This process is quantified by the
Precipitation Efficiency (PE) (e.g., Miltenberger, 2014; Dacre et al., 2023).

For the Lagrangian CR we obtain the following expression:

$$\text{CR} := \frac{C_{\text{hy}}(1) + E_{\text{v}}(1)}{\text{VAP}(1)}. \tag{7}$$

This formulation states that the condensation ratio is equal to the net hydrometeor growth by microphysical processes ($C_{\text{hy}}(1) + E_{\text{v}}(1)$) divided by the net initial water vapour content ($\text{VAP}(1)$). A detailed definition of the terms can be found in Section Appendix B4, but in short, $\text{VAP}(\tilde{t})$ is equal the initial vapour content minus the vapour lost to parameterisations other than cloud microphysics by the time $\tilde{t}$. CR thus gives the fraction of vapour initially present and carried in through the turbulence and
convection scheme (and numerical residuals) that is converted into hydrometeors. Using VAP, instead of $Q_m athrmtot(\tilde{t} = 0)$ as in the definition of DR, ensures that CR is bounded by 0 and 1.

For PE we obtain:

$$\text{PE} := \frac{P(1)}{C_{\text{hy}}(1) + E_{\text{v}}(1) + \text{HYD}(1)}. \tag{8}$$

This formulation states that PE is equal to the total net precipitation out of the parcel ($P(1)$) divided by the sum of net hydrometeor growth and net initial hydrometeor content ($\text{HYD}(1)$, initial hydrometeor content minus the hydrometeors lost to parameterisations other than cloud microphysics by the end of the ascent, see Appendix B4). In other words: PE tells us how many hydrometeors that i) were formed during the ascent, ii) were already present at the beginning, or iii) were carried in by the turbulence or convection scheme (or numerical residual) are precipitated out of the parcel by the end of the ascent. The
incorporation of iii) ensures that PE is bounded by 0 and 1.

Finally, we get $\text{DR}_{\text{mphys}}$ from PE and CR as follows:

$$\text{DR}_{\text{mphys}} = \text{PE}(\text{CR} + \epsilon)\frac{\text{VAP}(1)}{\text{VAP}(1) + \text{HYD}(1)}. \tag{9}$$

using

$$\epsilon := \frac{\text{HYD}(1)}{\text{VAP}(1)}. \tag{10}$$

We point out that:

$$\frac{\text{VAP}(1)}{\text{VAP}(1) + \text{HYD}(1)} = \frac{P(1)}{Q_{\text{tot}}(0) - Q_{\text{tcr}}(1)}, \tag{11}$$





is equal to one if all initial moisture is lost exclusively to precipitation. For small values of $\mathrm{HYD}(1)$ the fraction on the right hand side of eq. 9 goes to 1 and $\epsilon$ goes to zero, meaning that we can approximate:

$$\mathrm{DR_{mphys}} \approx \mathrm{PE} \cdot \mathrm{CR} \quad \text{for} \quad |\mathrm{HYD}(1)| \ll 1. \tag{12}$$

This equation is typically used in studies focussing on regional moisture budget analysis from an Eulerian perspective (e.g. Barstad et al., 2007; Miltenberger, 2014).

In this chapter we have presented all variables using the normalised ascent time scale introduced in Section 2.4. However, the definitions are also valid on the real time axis as long as the start and end times for the integrated rates are set accordingly. It is also possible to quantify at DR, PE, etc. at arbitrary times during the ascent by replacing $\tilde{t} = 1$ with a normalised time $\tilde{t} \in (0,1]$ (leaving the initial conditions unchanged). We will now discuss the physical meaning of PE and CR in $\mathrm{DR_{mphys}}$.

### 2.6.4 Interpretation of PE and CR's physical meaning in $\mathrm{DR_{mphys}}$

The physical meaning of PE and CR in $\mathrm{DR_{mphys}}$ can be easily interpreted if we assume that the net initial hydrometeor content $\mathrm{HYD}(1)$ is negligible. In this case $\mathrm{DR_{mphys}}$ approximates to $\mathrm{PE} \cdot \mathrm{CR}$. CR determines how much moisture is converted into hydrometeors and PE determines how many hydrometeors are removed. If CR is large and PE is small, then the air parcel is converting moisture efficiently but removing it inefficiently. Conversely, if CR is small and PE is large, then the parcel is efficient at hydrometeor removal but has low hydrometeor production. DR is less than one if some moisture remains, one if all moisture has been removed, and zero if no moisture has been removed. The following theoretical edge cases illustrate this:

1. $\mathrm{PE} = 1$, $\mathrm{CR} = 1$ and $\mathrm{DR} = 1$.

   There is no remaining moisture. All the vapour has been converted and every hydrometeor has precipitated.

2. $\mathrm{PE} = 1$, $\mathrm{CR} < 1$ and $\mathrm{DR} < 1$.

   Any remaining moisture is vapour and all hydrometeors have precipitated.

3. $\mathrm{PE} < 1$, $\mathrm{CR} = 1$ and $\mathrm{DR} < 1$.

   Hydrometeors make up all remaining moisture and there is no vapor.

4. $\mathrm{PE} = 0$, $\mathrm{CR} > 0$ and $\mathrm{DR} = 0$.

   Water vapour is converted into hydrometeors but they stay in the parcel. $Q_{\mathrm{tot}}$ is unchanged.

5. $\mathrm{PE} > 0$, $\mathrm{CR} = 0$, but $\mathrm{DR} > 0$ in the case that $\mathrm{HYD}(1) > 0$ is not negligible.

   This case only makes sense if the net initial hydrometeor content is not neglected, meaning $\epsilon > 0$. No moisture is converted and only the initial hydrometeors precipitate. $Q_{\mathrm{tot}}$ decreases but $Q_{\mathrm{v}}$ is unchanged.

As the last case illustrates, including $\mathrm{HYD}(1)$ ensures that initial hydrometeors and additional hydrometeor removal processes (e.g turbulence) are accounted for. Our interpretation of the role of PE and CR therefore holds as long as $\mathrm{HYD}(1)$ is much smaller than $\mathrm{VAP}(1)$. In Appendix Section B5 we show that our Lagrangian definitions for PE, CR and $\mathrm{DR_{mphys}}$ can heuristically be compared to the Eulerian definitions of the same variables from previous studies.



## 3 Case study

A strong cyclone with a pronounced WCB characterised the weather over the northern Atlantic from the afternoon of 22 September 2017 to the end of 23 September 2017. Measurements in air masses related to this system were carried out as part of the HALO-WISE campaign (Haynes and Palm, 2023). Note, it is an example of an almost purely open ocean WCB.

The conditions for cyclogenesis began to develop early on 20 September 2017. Strong southeasterly winds, resulting from
the combination of the windfields around a high pressure system off the coast of Newfoundland and a low pressure system off the southern tip of Greenland, drove a large and cold air mass from the Greenland and the Labrador Sea across the northern Atlantic (Figure A1 a). On the eastern side of the high-pressure system (located at 45°W, 45°N) warm air was transported northwards, leading to the formation of a 1500 km long front at 52°N (Figure A2 a). The converging winds associated with the high and low pressure systems acted frontogenically and strengthened the temperature gradient. This induced a baroclinic
zone with strong horizontal wind shear, and began lifting air masses at the point of convergence. At this time (ca. 04:00 UTC 21 September 2017), the low pressure system started forming.

At the same time, an upper-level trough/positive potential vorticity (PV) anomaly approached from the northwest and propagated to the southeast (Figure A2 b). It induced cyclonic rotation in the atmospheric column and combined with the lower-level
PV generation from latent heating formed a PV tower. This induced strong cyclonic flow in the lower levels of the atmosphere (Figure A3 a). As a result, the cyclone underwent explosive cyclogenesis, with central surface pressure dropping from 1002 hPa at 04:00 UTC 21 September to 977 hPa at 04:00 UTC the following day. In the cloud field a comma shaped cloud band typical for WCBs formed on 22 September 2017 and propagated northeast until its northern tip reached Iceland early on 23 September (Figure 1). At this point the cold front, along which convective clusters were found, stretched 4000 km across the Atlantic
and reaching south of 40°N. On 24 September 2017 the cyclone dissipated over the Norwegian Sea. The upper level trough propagated further downstream (Figure A5 b). On the surface the cyclone brought relatively mild air (ca. 15 °C) all the way to Iceland and Svalbard and therefore had a large impact on central and northern European weather.

From our simulation of the case (see section 2.2 for a description of the set-up) we identified 393 070 WCB trajectories
associated with the cyclone described above. The spatial and pressure evolution for every tenth trajectory throughout the entire simulation is shown in Figure 3. The coloring shows that most trajectories experience a large change in pressure (and hence their WCB ascent) within the highest resolution domain. The average horizontal distance covered during the ascent is 2400 km. A few trajectories rise very early and remain in the southern region of the WCB. The majority of trajectories move along northwards during their ascent, inline with the comma-shaped cloud structure, and move eastwards after reaching the UTLS. Some
then turn clockwise and move to the south-east, reaching as far south as Morocco, while others, completing their ascent further north, move across central or northern Europe. A large proportion end up as far north as Greenland.





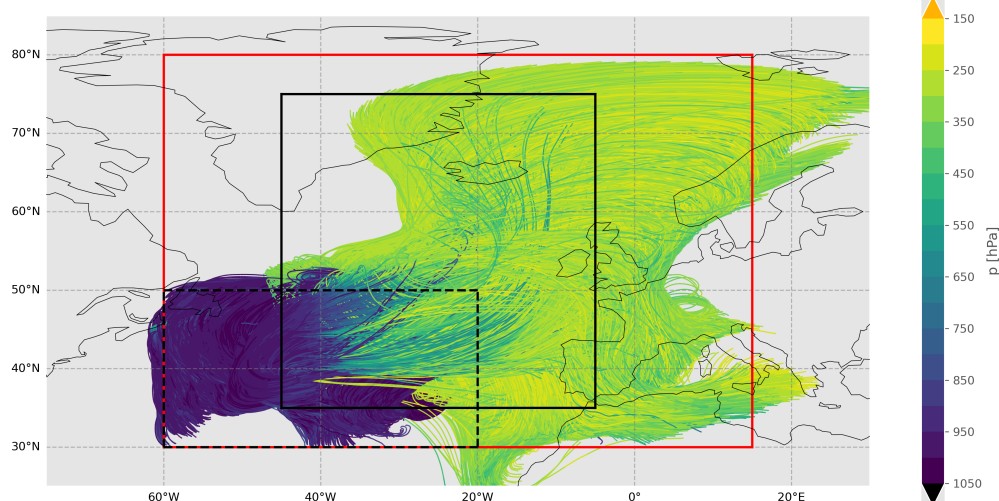

**Figure 3.** The geographical and pressure course of some selected WCB trajectories for the entire simulation time in relation to the first nest (red), second nest (black) and starting area (black dashed).

## 4 Ascent statistics and convective behaviour

We characterise the ascent of WCB trajectories on the basis of their $\tau_{600}$ (section 2.3). Figure 4 a shows a histogram of $\tau_{600}$ and $\tau_{\mathrm{WCB}}$ ascent times for all selected WCB trajectories. We see a uni-modal distribution with mean, median and mode for 345   $\tau_{600}$ are 16.9, 14.5 and 11 h, respectively. A large variability of ascent timescales with a steep decline for timescales shorter than about 10 h and a tail towards longer ascent times. We define three ascent "regimes": (i) "convective" trajectories with $\tau_{600} < 5\,\mathrm{h}$, representing 6% of all WCB trajectories, (ii) "normal" trajectories with $18\,\mathrm{h} \geq \tau_{600} \geq 6\,\mathrm{h}$ representing 55%, and (iii) "slow" trajectories with $\tau_{600} > 20\,\mathrm{h}$, representing 30% of all cases. However, $\tau_{600}$ ascent times do not form a complete picture of ascent behaviour, as they quantify only deep ascent. The 300 hPa and 400 hPa ascent times ($\tau_{300}$, $\tau_{400}$, Figure 4 b) 350   focus on smaller-scale embedded convective activity. There distribution is not unimodal, but instead bimodal with peaks at about 5 h (3 h) and 1.5 h (1 h) for $\tau_{400}$ ($\tau_{300}$). Many trajectories therefore ascend rapidly for a large part of the total 600 hPa ascent, but fail to maintain a high rate of ascent throughout the troposphere (an example for this is shown in Figure C1 b).

The omnipresence of rapid ascent segments across the $\tau_{600}$ spectrum is corroborated by the distribution of maximum pres- 355   sure drop in 2 h (Figure C1,a): 38% of slow trajectories experience periods where they rise about 200 hPa in 2 h. 78% of all trajectories have a 0.5 h period in which the average ascent velocity is larger than 120 hPa/h (gives 600 hPa in 5 h). Many trajectories also rise in "steps", with two or more sections of rapid ascent followed by extended periods of almost no ascent. 30% of slow trajectories experience a 10 h period in which they ascend less than 10 hPa, 10% even *descend* more than 40 hPa (not shown). Only 9% of slow trajectories have $\max(\Delta p_{2\,\mathrm{h}}) < 100\,\mathrm{hPa}$. Truly slantwise ascending trajectories are therefore in the 360   minority and one cannot assume that a large $\tau_{600}$ value means that the entire ascent is slow and monotonous.





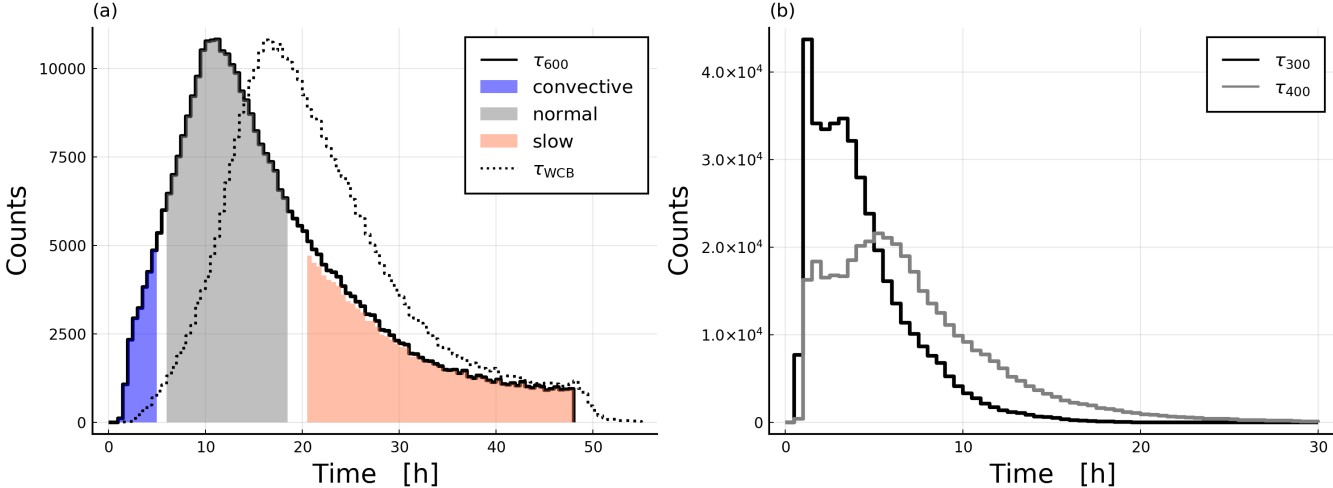

**Figure 4.** Ascent statistics that help us understand the convective behaviour of WCB trajectories. In a) the $\tau_{600}$ and $\tau_{\mathrm{WCB}}$ ascent times are shown, as well as the three categories into which they are grouped. In b) the $\tau_{300}$ (black) and $\tau_{400}$ (grey) ascent times are shown. The distributions are different, because many trajectories ascend a large portion convectively but not the full $600\,\mathrm{hPa}$.

To ensure that the problematic cases (where $\tau_{600}$ is large but the trajectory is rising convectively for most of the ascent) are neglected, we remove slow trajectoriesthat rise quickly for a large part of the ascent, i.e. $\max(\Delta p_{2\,\mathrm{h}}) < 350\,\mathrm{hPa}$, from the data-set for further analysis (removes about 5 % of slow trajectories). A similar condition has been imposed by Rasp et al., 365  2016. This additional condition is why the orange color in Figure 4 a does not fill the histogram all the way to the top.

In summary, the WCB trajectories show a wide range of ascent timescales and almost all of them, even the slowly ascending ones, experience bursts of convective activity. This shows that even for an open ocean WCB, convection is a ubiquitous feature, and that truly "slantwise" ascending trajectories are a minority. We will now analyse the transport of moisture by the 370  WCB.

## 5   Moisture transport analysis

WCB ascent connects the planetary boundary layer with the upper troposphere and thereby constitutes a source of moisture for the UTLS. The actual amount of water reaching the UTLS depends on the initial water content at the start of the ascent and loss process during the ascent. For the selected case study, we present a detailed analysis of this moisture budget in the following.



## 5.1 Trajectory characteristics at the start of the WCB ascent

The conditions at the start of the ascent determine the maximum amount of moisture that WCB trajectories can transport, if loss would occur during ascent. Convective trajectories have initial temperatures and pressures of approximately 20 °C and 975 hPa, respectively, whereas slow trajectories have about 18 °C and 965 hPa (Figure 5 a, recall that $\tilde{t} = 0$ is the start of the $\tau_{\mathrm{WCB}}$ ascent). Hence, on average faster ascending trajectories begin their ascent at higher temperatures and pressures and have a higher initial total moisture content than slower ascending trajectories. However, the spread of conditions for the start of slow trajectories is much larger than for the fast trajectories, as can be seen by the difference in the mean and median values. This is consistent with the finding by Oertel et al. (2023) that convective trajectories originate in a more southerly part further away from the cyclone center compared to slow trajectories. The initial total water content behaves similar to the temperature, i.e. it is decreasing with increasing ascent time (Figure 6 a). On average, $Q_{\mathrm{tot}}(0)$ is 12.9 gkg$^{-1}$ for convective trajectories and 11.3 gkg$^{-1}$ slow trajectories. The initial hydrometeor content varies is generally small (<0.4 g/kg, average of 0.08 g/kg for all trajectories, Figure 6 a) with a weak dependence on $\tau_{600}$. Also for the moisture variables the spread increases with ascent time. This means that we find a strong correlation of short ascent times with high vapour content and high temperatures. We suggest that this is because higher temperatures and vapour content mean that more buoyancy can be generated by latent heating, leading to faster ascent times.

## 5.2 Trajectory characteristics at the end of the WCB ascent

Before investigating the processes by which moisture is lost during ascent, we now characterise the thermodynamic conditions at the end of ascent and the moisture content of WCB trajectories.

**Temperature and pressure**

The pressure and temperature at the end of ascent is lower for trajectories with smaller $\tau_{600}$ values compared to more slowly rising trajectories (recall that $t = 1$ is the end of the $\tau_{\mathrm{WCB}}$ ascent) (Figure 5 b). This difference persists even many hours after completion of the ascent, with the minimum pressure and temperature that convective trajectories reach throughout the entire simulation averaging 236 hPa and -56 °C, and for slow trajectories 314 hPa and -44 °C (not shown). This is partially due to the fact that faster trajectories ascend earlier and therefore further south (not shown), where the atmosphere is deeper. However, even when trajectories are rising at the same latitude, those with lower $\tau_{600}$ values will rise on average 90 hPa higher. The difference is therefore more due to differences in initial humidity and temperature, where larger values favor latent heat release and stronger cross-isentropic transport.

**Hydrometeor content**

Convective trajectories transport far more hydrometeors (mass and number) into the UTLS than slow trajectories (Figure 6 b). Although convective trajectories make up only 5.8% of all trajectories, they account for 14.3% of the total hydrometeor mass of all trajectories at $\tilde{t} = 1$. The mean for $Q_{\mathrm{hy}}(1)$ remains below 0.04 g/kg for $\tau_{600} > 10$ h and the spread is low. $Q_{\mathrm{hy}}$ strongly





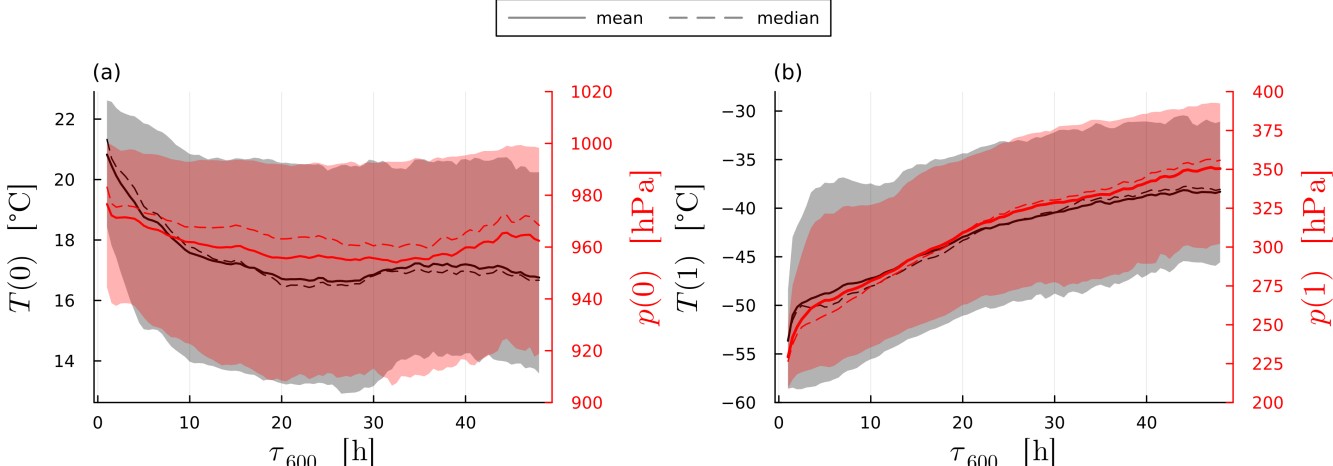

**Figure 5.** Mean (solid line) and median (dashed line) of the temperatures (black) and pressures (red) at the beginning (a) and end (b) of the WCB ascent. 10 th and 90 th percentiles are shaded. Faster trajectories start their ascent at higher temperatures and pressures, and conversely end their ascent at significantly lower temperatures and pressures. Note: the pressure axis is NOT flipped to make the correlation with temperature more obvious.

increases for shorter $\tau_{600}$ reaching a maximum value of $Q_{\mathrm{hy}}(1) = 0.17\,\mathrm{g/kg}$ is reached $\tau_{600} = 1.5\,\mathrm{h}$. Note that for all $\tau_{600}$ the total hydrometeor content at the end of the ascent ($Q_{\mathrm{hy}}(1)$) is roughly 99% ice and 1% snow. No liquid hydrometeors are found although T(1) is not for all trajectories below 235 K. This shows that the hydrometeor content at the end of the ascent is strongly influenced by ascent time.

**Vapour content and relative humidity over ice**

Aside from condensate, WCB trajectories also transport water vapor to the UTLS. Most trajectories have vapour contents that are thermodynamically constrained, which can be seen by the correlation of specific humidity with the saturation specific humidity over ice calculated using the temperature and pressure (Figure 7 b). Many convective and slow trajectories therefore have a vapour content that is at or slightly below saturation with respect to ice (relative humidity over ice ($\mathrm{RH}_{\mathrm{i}}(1)$) shown in Figure 7 a). Sub-saturated conditions likely arise due to the trajectories at the edge of the WCB outflow region or some subsidence being incorporated into the last 30 min output interval classed as end of WCB ascent. On average, convective trajectories have an $\mathrm{RH}_{\mathrm{i}}(1)$ of 102.6% compared to $\mathrm{RH}_{\mathrm{i}}(1)$ of 104.7% for slow trajectories (104.7%). Distinct difference are seen in the distribution of $\mathrm{RH}_{\mathrm{i}}(1)$ (Figure 7 a), which shows a sharp peak at 97% and a tail to larger values, whereas the distribution for slow trajectories is more bell-shaped with a maximum at 108%. The result is that, although the average is similar, only 56.5% of convective trajectories are supersaturated compared to 80% of slow trajectories. This is an interesting finding given that convective trajectories have a much higher hydrometeor content than slow trajectories at the end of the ascent. However, the number concentration of ice in slow trajectories is a smaller (Figure 11, which likely impact the supersaturation relaxation



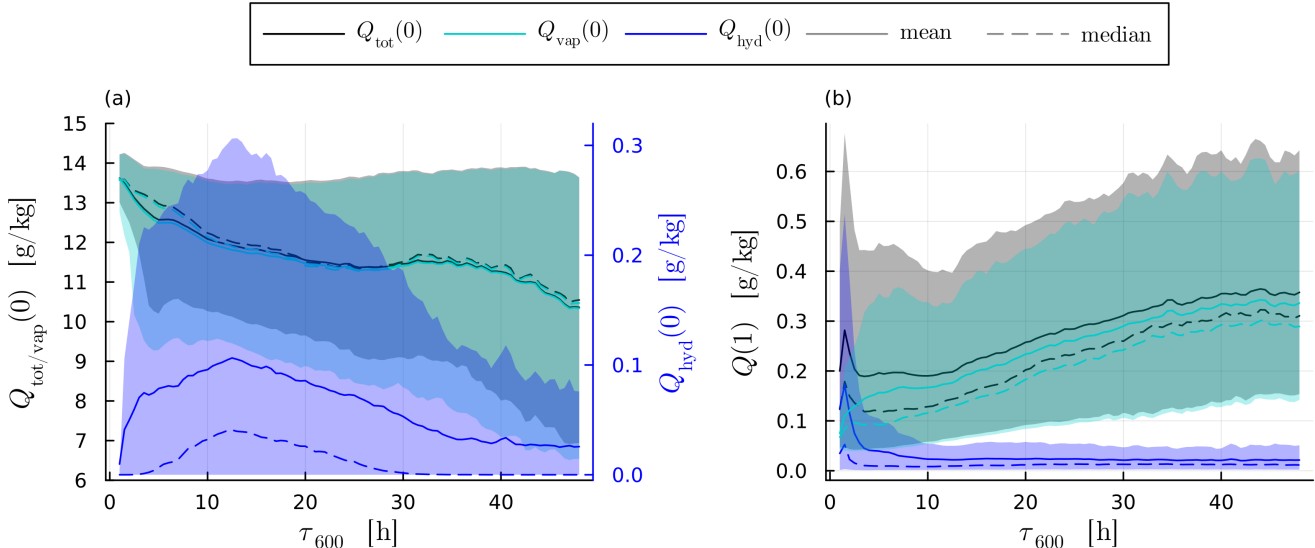

**Figure 6.** Mean (solid line) and median (dashed line) total moisture content ($Q_{\text{tot}}$, black) as well as the vapour ($Q_{\text{vap}}$, cyan) and hydrometeor ($Q_{\text{hyd}}$, blue) content at the beginning (a) and at the end (b) of the WCB ascent. 10 th and 90 th percentiles are shaded. Note the different y-axis for $Q_{\text{hyd}}(0)$ in (a) due to the difference in magnitude.

timescale. Across all ascent timescales, we find that 70% are supersaturated with respect to ice when they complete their ascent and enter the UTLS. This means that the WCB introduces large amounts of ice-supersaturated air into the UTLS.

If we consider the final vapour content $Q_v(1)$, we find a strong dependence on $\tau_{600}$ with increasing specific humidity for
slower ascending trajectories that cannot be only explained by the differences in $\text{RH}_i$ (Figure 6). This overall trend is rather explained by the different outflow temperatures (Figure 5), as evident from a tight correlation of $Q_v(1)$ with the expected saturation mixing ratio (Figure 7). The scatter away for the correlation is related to variability in the $\text{RH}_i$ and consistent with discussion in the previous paragraph increases towards larger $Q_v(1)$ values, which are associated with warm outflow temperatures and large $\tau_{600}$.


We can summarise this section by noting that convective trajectories reach higher into the atmosphere at the end of the ascent, have lower specific humidity than slow trajectories and higher hydrometeor contents than slow trajectories. The specific humidity at the end of ascent is to the first order thermodynamically constrained by the outflow temperature. Substantial spread in the specific humidity at a given temperature is introduced by deviations away from saturation, which are more abundant
in slow compared to convective trajectories with an $\text{RH}_i(1) \geq 1$ for 80 % and 56.5 % trajectories, respectively. At the end of ascent convective trajectories contain on average a factor 10 (median factor 2) larger ice crystal number concentration and about twice as much ice mass. Hence, while fast convective and slow trajectories transport a similar amount of total water mass



to the UTLS (with a minimum for intermediate ascent timescales), the partitioning between water vapor and condensate as well as the ice cloud properties depend strongly on the ascent timescale.

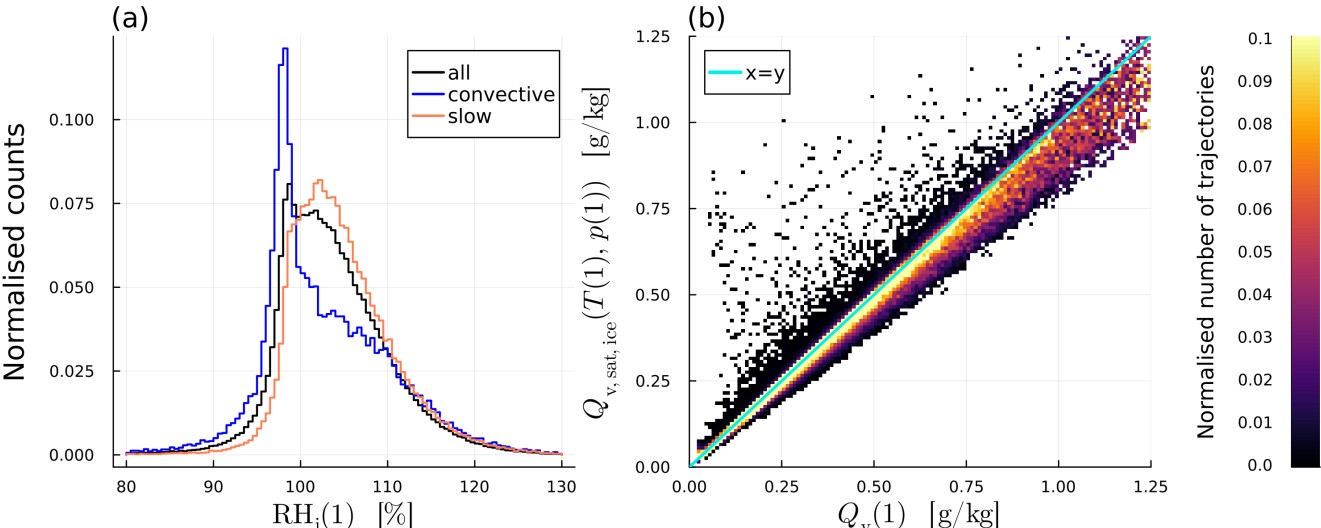

**Figure 7.** (a) Relative humidity over ice for all trajectories as well as the convective and slow trajectories, shown as a normalised histogram at the end of ascent or all (black), convective (blue) and slow (orange) trajectories. (b) Calculated saturation specific humidity (over ice) given temperature and pressure at the end of the ascent, over the specific humidity.

### 5.3 Moisture loss pathways

Having considered the differences between convective and slow trajectories in terms of their initial and final conditions, and the evolution of variables after ascent, we will now compare the behaviour of moisture loss with differing ascent times. The variables introduced in Section 2.6 characterise and quantify the loss of moisture by WCB trajectories. In this section we investigate the behaviour of these variables with $\tau_{600}$.

**Total fractional moisture loss** (DR)

DR indicates the percentage of moisture lost by the end of the ascent, and on average WCB trajectories lose more than 95% of their moisture (Figure 8 a). DR decreases with increasing ascent time, meaning that faster trajectories lose a larger fraction of their initial moisture than slower trajectories. As we will discuss in the next paragraphs, this is mainly due to the lower temperature attained by fast ascending trajectories, which results in low saturation vapor pressure (see discussion on CR). The slight decrease in DR for the fastest ascending trajectories is due to their elevated hydrometeor content (Figure 6 a, see discussion on PE). This shows that DR is strongly controlled by the thermodynamics and modulated by the transport of hydrometeors.



**Fractional moisture loss due to mixing processes** ($\mathrm{DR_{mix}}$)

The fraction of the initial humidity that is removed by turbulent mixing during the ascent is quantified by $\mathrm{DR_{mix}}$. Overall, $\mathrm{DR_{mix}}$ increases with increasing ascent time (Figure 8 c), meaning that slower ascending trajectories experience more fractional moisture loss due to processes other than precipitation than fast ascending trajectories. The individual contributions to $\mathrm{DR_{mix}}$ (turbulence, convection parameterisation, numerical residuals) as functions of $\tau_{600}$ are shown in in Figure C4. The

largest contribution to $\mathrm{DR_{mix}}$ comes from the turbulence parameterisation; the convection parameterisation and numerical residuals play a secondary role. This makes sense because a longer ascent time means a longer time for surrounding air to be mixed into the air parcel through turbulence. However, for the longest ascent times $\mathrm{DR_{mix}}$ decreases slightly. We presume that this is because the slowest trajectories move along regions that experience smaller horizontal and vertical wind shear (which drive turbulence). Circumstantial evidence is provided by temporally averaged wind shear amplitudes along the ascent (Figure

C2). This indicates that slower ascending trajectories are more part of the large scale coherent flow of the WCB than fast ascending trajectories. For all WCB trajectories the moisture loss due to mixing occurs in the first part of the ascent at pressures larger than about 700 hPa (Fig. C5 a and b).

**Fractional moisture loss due to precipitation** ($P(1)/Q_{\mathrm{tot}}(0)$)

The net precipitation $P(1)$ decreases with increasing ascent time (Figure C3 a) correlating with less available moisture in the slowly ascending trajectories (Figure 6 a). This suggests (and is supported by precipitation efficiency metrics discussed later) that precipitation formation is very efficient and precipitation amounts are strongly controlled by thermodynamic constraints on condensate formation.

The dominant form of net precipitation for almost all trajectories is warm phase (rain) (Figure C3 a). Only the fastest ascending trajectories have equal amounts of frozen precipitation (ice, snow, graupel, hail) and rain. The net flux of frozen hydrometeors is negative for many slow ascending trajectories, and is largely a mirror image of the net rain flux. This suggests that slow trajectories convert frozen hydrometeors entering them from above (mainly graupel and snow, Figure C3 b) into rain by melting. As discussed later slow trajectories spend a large fraction of their ascent just below the melting layer, which allows

for substantial influx of frozen hydrometeors in melting conditions (section 5.4). For all trajectories with $\tau_{600} < 20\,\mathrm{h}$, graupel precipitation is the dominant precipitating hydrometeor and makes up 33% of precipitation for convective trajectories (Figure C3 b). So for all trajectories, but especially for fast ascending trajectories, frozen hydrometeors play a larger role in removing moisture.

Fast ascending trajectories experience greater fractional moisture loss due to net precipitation ($P(1)/Q_{\mathrm{tot}}(0)$) than slow ascending trajectories (Figure 8 f). $P(1)/Q_{\mathrm{tot}}(0)$ is larger than $\mathrm{DR_{mix}}$ for all trajectories, showing that precipitation is the dominant mechanism for moisture loss of WCB trajectories, regardless of their ascent time. Note that $P(1)/Q_{\mathrm{tot}}(0)$ is essentially the mirror image of $\mathrm{DR_{mix}}$ as the two terms contain all possible moisture loss processes.



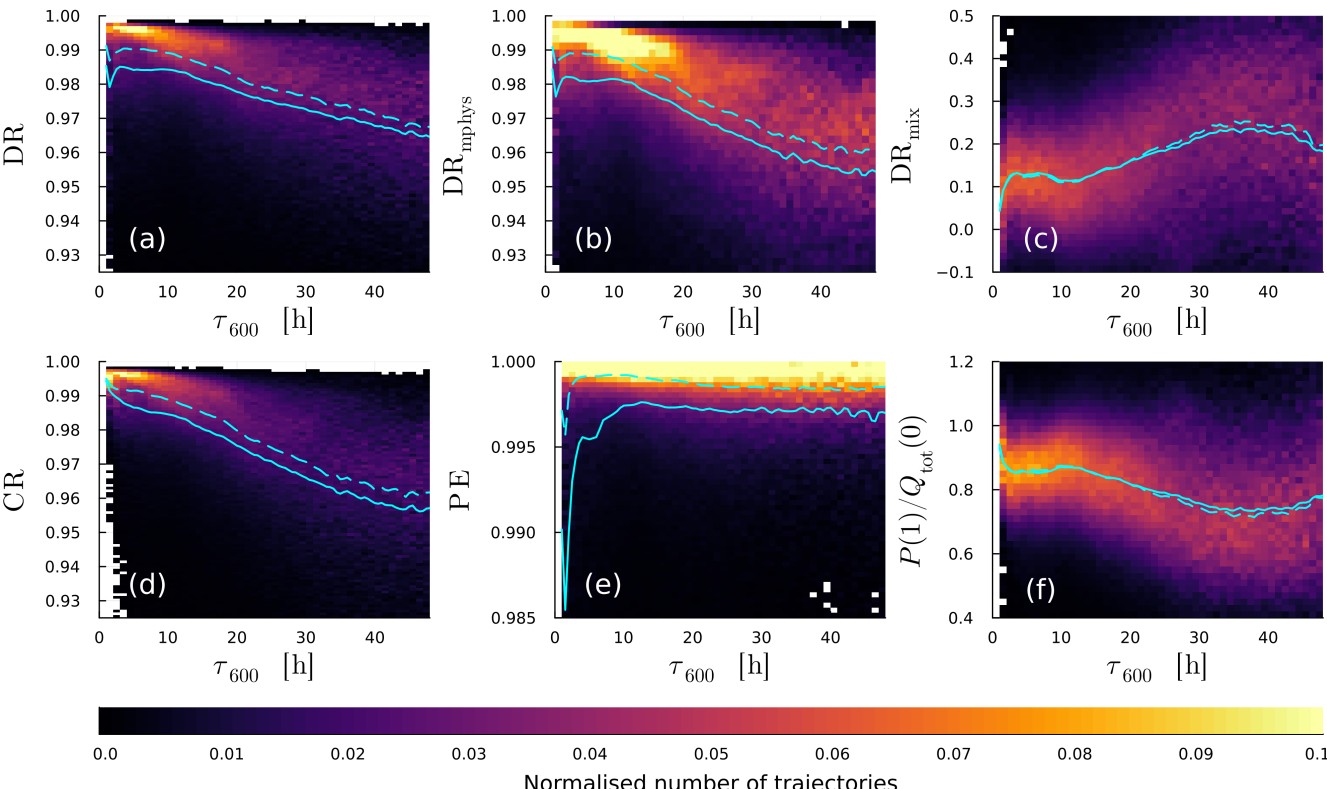

**Figure 8.** Two dimensional normalised histograms showing DR (a), $\mathrm{DR_{mphys}}$ (b), $\mathrm{DR_{mix}}$ (c), CR (d), PE (e) and $P(1)$ (f) over $\tau_{600}$. Mean (solid) and median (dashed) are plotted in cyan.

**Efficiency of moisture loss through precipitation** ($\mathrm{DR_{mphys}}$, PE, CR)

Finally, we quantify the relative role of condensate formation and sedimentational removal of condensate for shaping the variability of net precipitation and outflow moisture content with ascent timescale $\tau_{600}$. The fraction of inflow moisture that is not removed by turbulence and is converted to condensate is quantified by the Lagrangian condensation ratio CR. CR is on average 0.98 and decreases from about 0.99 for convective to about 0.97 for slowly rising trajectories (Figure 8 d). The large CR reflects the large vertical displacement of all trajectories and facilitates the strong cross-isentropic ascent of the WCB. Water vapour content at the end of the ascent is close to saturation for most trajectories (section 5.2) and is therefore to first order thermodynamically constrained. Hence it is not surprising that the dependency of CR on $\tau_{600}$ reflects the outflow temperature of the WCB trajectories.

The fraction of the condensate that is removed by gravitational settling can be quantified by the Lagrangian precipitation efficiency PE. PE is very close to one for all trajectories (Figure 8 e) indicating that WCB air parcels are extremely efficient at removing the hydrometeors they form/grow during the ascent. For the fastest ascents, PE decreases sharply, which is reflected




in the large final hydrometeor content for convective trajectories. Because of the longevity of ice particles in convective trajectories, PE also remains larger for faster ascents times in the hours after the ascent (not shown).


Finally, PE and CR can be combined to a microphysical drying ratio $DR_{mphy}$ analogous to the $DR_{mix}$ and the fractional moisture loss discussed earlier. In contrast to the fractional moisture loss, $DR_{mphy}$ quantifies the loss of moisture due to cloud microphysical processes in the absense of turbulent mixing. $DR_{mphys}$ is approximately equal to $PE \cdot CR$ as HYD(1) and $\epsilon$ are much smaller than one (not shown). Further as PE is essentially 1 for most trajectories, $DR_{mphys}$ is almost equal to CR. Only

for the fastest trajectories do we see a dip in $DR_{mphys}$ that reflects the sharp decrease in PE.

In summary, the main pathway for moisture removal for all trajectories is precipitation with mixing processes playing a secondary role. Fractional moisture loss is >95% for all trajectories and is mainly controlled by the vertical displacement that WCB parcels experience, i.e. temperature at the end of the ascent. This is reflected in CR, which behaves like to final pressure and temperature, and PE, which is close to 1. The exception for PE are convective trajectories, where PE decreases sharply

with decreasing ascent time. This shows that convective activity has a large impact on the efficiency with which moisture is removed by the end of the WCB ascent. Note that the variability of PE, CR, and $DR_{mpyhs}$ with $\tau_{600}$ may appear small because all parcels are rising very strongly, but this variability gives rise to substantial variability in the moisture content in the UTLS (about a factor of 2 in $Q_{tot}(1)$, section 5.2).

**5.4   Trajectory characteristics during the ascent**

Here we present detailed analysis of the moisture and thermodynamic evolution of rising WCB air parcels to provide a physical insight into the processes that produce the $\tau_{600}$ dependence of moisture budget terms. In particular, we contrast slowly and convectively rising parcels and display their evolution on an normalised ascent-time axis as discussed in section 2.4. We focus on the microphysical removal of moisture and the respective terms of the moisture budget, i.e. CR and PE.


**Processes contributing to condensation efficiency** CR

The condensation ratio CR increases with decreasing $\tau_{600}$, as discussed in section 5.3. The investigation of end of ascent moisture content showed a strong impact of outflow temperature on absolute specific humidity. This is confirmed by an analysis of the time evolution of CR (Fig. 9 a) and temperature (Fig. 9 b): CR decreases much faster for convective trajectories than for

slow ones mimicking the temperature evolution. CR of 80 % is reached at a temperature of $-20$ C for slow trajectories and slightly warmer temperatures for convective trajectories.

In section 5.3 we further found distinct difference in the relative humidity distribution, i.e. deviations away from the thermodynamic equilibrium specific humidity, which modulate the temperature control. To elucidate the physical processes resulting in

the different relative humidity distributions, we show the time evolution of relative humidity over water $RH_w$ and ice $RH_i$, of the liquid fraction, i.e. the ratio of liquid condensate mass to total condensate mass, and the ice crystal number concentration





**Figure 9.** Condensation ratio $\mathrm{CR}$ (a) and temperature (b) over normalised ascent time. Relative humidity over water $\mathrm{RH_w}$(c), relative humidity over ice $\mathrm{RH_i}$ (d), liquid fraction (e), and ice crystal number concentration (f) over parcel temperature during the WCB ascent. In all panels the mean (solid) and median (dashed) as well as the 10th and 90th percentiles (shaded) are plotted in blue for convective and orange for slow trajectories.

in Fig. 9 c-f. Note that the abscissa is now temperature, which allows us to focus on deviations away from thermodynamic equilibrium. The median $\mathrm{RH_w}$ increases at the warm temperatures as WCB parcels approach the cloud base and reaches close to 100 % at about $12\,^\circ\mathrm{C}$ for both sets of trajectories. It remains close to 100 % for a substantial part of the ascent reflecting

the control of saturation adjustment on the specific humidity in the model. Note that small deviations away from 100 % are likely due to interpolating specific humidity, temperature and pressure independently to the trajectory position and computing the relative humidity from the interpolated values.





For slow trajectories the median $\mathrm{RH_w}$ decreases rapidly for temperatures below about $-6\,^\circ\mathrm{C}$, while conditions close to water saturation persist to temperatures of about $-17\,^\circ\mathrm{C}$ for convective trajectories. This difference is due to the more rapid glaciation of slowly rising trajectories compared to convective ones, as is also evident from the larger liquid fraction of around 0.25 below $-10\,^\circ\mathrm{C}$. Smaller vertical velocities (Fig. C5 c) mean that both the Wegener-Bergeron-Findeisen process is more efficient and that there is a larger ratio of the available growth time to the timescale of depositional growth in slow trajectories (Fig. C6 a-c). Note that 90 % of convective and slow trajectories are fully glaciated before reaching the homogeneous freezing temperature for cloud droplets ($-38\,^\circ\mathrm{C}$). Consistent with this evolution, $\mathrm{RH_i}$ rapidly drops to values close to 100 % only after the liquid fraction decreases below about 10-20 %. In the ice-phase part of the ascent (temperatures between $-30\,^\circ\mathrm{C}$ and $-40\,^\circ\mathrm{C}$, the median $\mathrm{RH_i}$ is smaller for convective compared to slow trajectories despite larger vertical velocities. The likely reason is an about a factor two ice crystal number concentration in the convective trajectories in this temperature range (Fig. 9 f). In the Hande et al. (2015) parameterisation less INP are available for deposition nucleation compared to immersion freezing in the relevant temperature range. As glaciation occurs at warmer temperatures in the slow trajectories, immersion freezing will generate less ice crystals in slow compared to convective trajectories. Together with the shorter time available for sedimentation and aggregation of ice crystals to snow this likely explains the difference in ice crystal number concentration, as heterogeneous ice nucleation in the model is purely temperature dependent.

In summary, the dependence of CR and the $\mathrm{RH_i}(\tilde{t}=1)$ distribution on the ascent timescale $\tau_{600}$ can be explained by the stronger vertical displacement and less rapid glaciation in convective trajectories. The use of saturation adjustment as well as the details in the ice nucleation parameterisation may impact the quantitative results, which the impact of which we will explore in a future study.

**Processes contributing to precipitation efficiency** PE

The precipitation efficiency PE also increases with decreasing $\tau_{600}$, with a weak indication of reduced PE for the fastest ascents ($\tau_{600} \leq 2\,h$) as discussed in section 5.3. Considering the temporal evolution of PE and relevant microphysical properties during WCB ascent once more reveals the underlying physical processes. Fig. 10 a shows the time evolution (in normalised ascent time $\tilde{t}$) of PE. Note that the value of at a given time corresponds to the fraction of condensate formed up to this point that has already been removed from the considered air parcel. The PE evolution is much more similar between slow and convective trajectories than the evolution of CR and not tied to the very different temperature and pressure history. This further corroborates the hypothesis that CR and PE are useful metrics for understanding (thermo-)dynamic and microphysical controls on precipitation formation (see also dicussion in e.g. Miltenberger et al., 2016; Miltenberger, 2014). PE increases very quickly to 0.9 in the first quarter of the ascent corresponding roughly to the evolution below 800 hPa (Fig. C5 b). Microphysics is dominated by warm-rain processes (Fig. 9 b) and accretion and parcels move to below the main WCB cloud, i.e. are increasingly affected by hydrometeor influx from above (not shown). Consistently the hydrometeor mass mixing ratio strongly increases in the first quarter of the ascent along slow and convective trajectories (Fig. 10 c) suggesting that condensate loss is less efficient than condensate production. At the end of this first phase of the ascent both sets of trajectories have a similar PE of about 0.9.





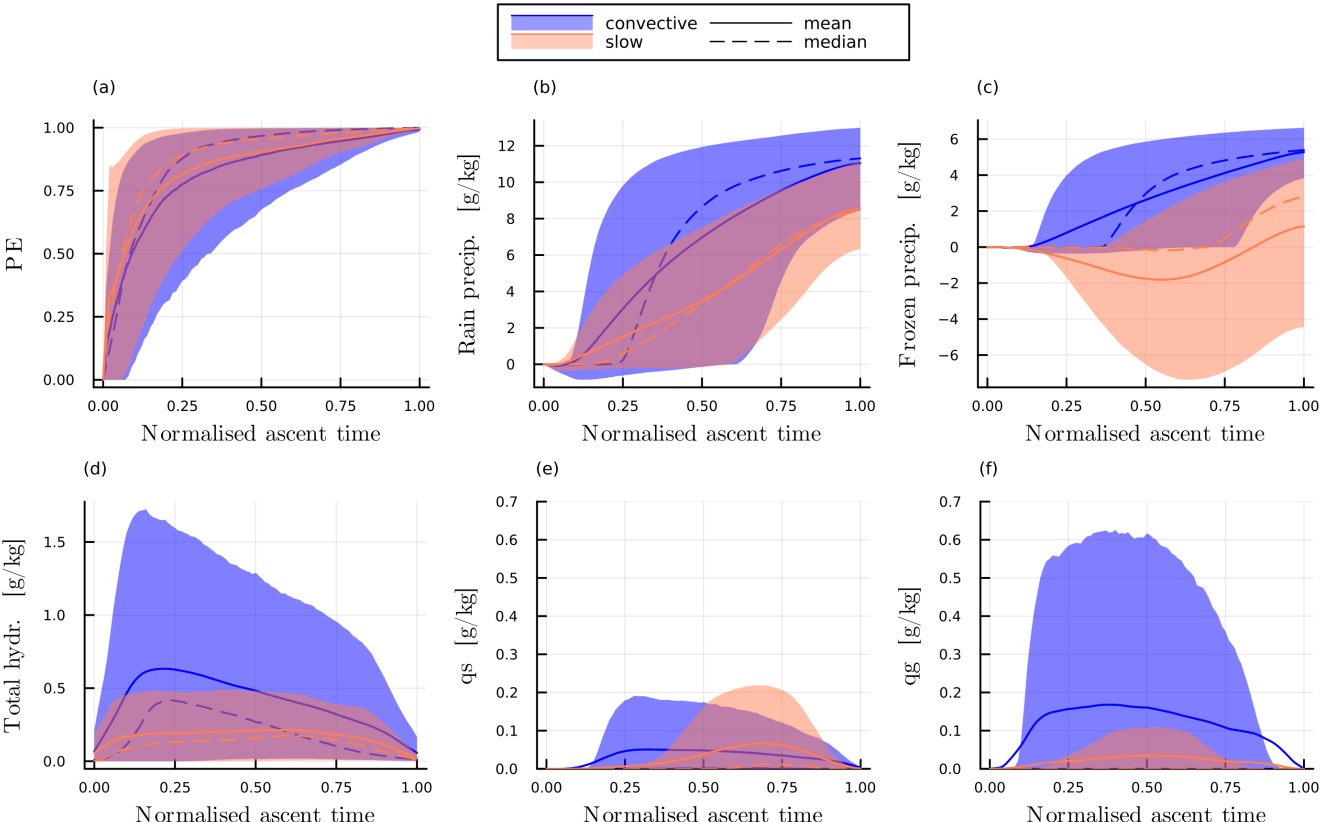

**Figure 10.** Precipitation efficiency PE (a), net rain precipitation (b), net frozen precipitation (c), as well as total hydrometeor (d), snow (e) and graupel (f) mass mixing ratio over normalised ascent time. In all panels the mean (solid) and median (dashed) as well as the 10th and 90th percentiles (shaded) are plotted in blue for convective and orange for slow trajectories. Note that a positive net precipitation signifies a loss of condensate from the trajectories from gravitational settling.

In the evolution beyond $\tilde{t} = 0.25$, PE slowly increases for both sets of trajectories and by the end of the ascent is slightly larger for convective trajectories. Given an almost identical median PE at $\tilde{t} = 0.25$ the microphysical evolution at later times is decisive. Interestingly, the median loss of water by rain sedimentation is almost identical for slow and convective trajectories albeit the distribution for slow trajectories being strongly skewed towards large values (Fig. 10 b). This tail towards large loss by rain is likely due to the longer residence of parcels in the vicinity of the melting layer (up until $\tilde{t} = 0.5$, Fig. 9 b). This is

supported by the mean negative net frozen precipitation for slow trajectories (Fig. 10 c), which implies frozen precipitation is entering the parcel, melting in the parcel and leaving as rain. The melting will impact the latent heating history of air parcel (consistent with stronger latent cooling from melting found for slow trajectories in Oertel et al. (2023)), but does not impact the moisture removal directly. Therefore warm-phase microphysics are explaining the $\tau_{600}$ dependence of PE.



Beyond $\tilde{t} = 0.25$ the hydrometeor mass mixing ratio steadily decreases for convective trajectories, while it remains almost constant or even slightly increases for slow trajectories (Fig. 10 b). This implies that given a similar rate of condensate formation, efficiency of precipitation formation increases more strongly in convective compared to slow trajectories. Interestingly, the median frozen precipitation signifies no net loss of condensate mass up until $\tilde{t} = 0.75$ for slow trajectories, while median net frozen precipitation is rapidly increasing beyond about $\tilde{t} = 0.4$ for convective trajectories (Fig. 10 c). This implies that for

slow trajectories, frozen precipitation falling into the parcel is not collecting a lot of condensate. In contrast frozen precipitation falling into convective trajectories collects a lot of condensate amplifying the sedimentation flux across the air parcel. Considering partitioning of frozen condensate into different types shows that convective trajectories have a much larger share of graupel, while snow formation dominates frozen precipitation in slow trajectories (Fig. **??** e and f). Larger graupel mass mixing ratios for graupel are consistent with the larger vertical velocities (Fig. SX), larger liquid fraction (Fig. 9 e), and a more vertical

compared to slantwise ascent (not shown) of convective trajectories. Large graupel mass mixing ratios also suggest efficient riming growth (Fig. C6 c), which explains positive net frozen precipitation. Towards the end of ascent and after full glaciation, i.e. $\tilde{t} \geq 0.75$ also slow trajectories have a positive net frozen precipitation, which is predominantly snow (not shown).

In summary, the $\tau_{600}$ dependence of PE is due to different mixed-phase precipitation formation pathways controlled by vertical

velocities and availability of supercooled liquid: riming growth dominates precipitation formation in convective trajectories, while aggregation and deposition is more important for hydrometeor growth in slow trajectories. This is consistent with findings in earlier studies on precipitation formation. The difference in precipitation formation pathway as well as associated mass growth rates and particle fall velocities result in decreasing PE for slow trajectories despite the longer time available for precipitation formation and sedimentation. For very short $\tau_{600}$ there is an indication of reduced PE. We refrain from further analysis,

as there are only very few trajectories in this category and the output time resolution of our trajectory data is too coarse to allow for a meaningful analysis.

## 5.5 Trajectory characteristics after the ascent

Since virtually all hydrometeors at the end of the ascent are ice, the differences in hydrometeor contents in the hours after the ascent between convective and slow trajectories are also seen when only looking at ice particle mass and number concentrations (Figure 11). A particularly large difference is seen in the ice *number* concentration, with convective trajectories containing up to 15 times more ice particles than slow trajectories directly at the end of the ascent. This difference remains large up to 24 h after the ascent, after which convective trajectories contain (on average) roughly $10^4 \ kg^{-1}$ ice particles and

slow trajectories roughly $5 \cdot 10^3 \ kg^{-1}$ ice particles (Figure 11 a). Ice mass mixing ratios for both groups are similar roughly 5 h after the end of ascent (Figure 11 b). The initial strong decrease in number concentration and particle mass is due to sedimentation and aggregation of (large) ice crystals. This is reflected in the geometric radius of ice particles (Figure 11 c), which





decreases with time. There is a small increase/stagnation of ice particle radius after 5 h, which is due to the production of new ice particles through homogeneous and heterogeneous freezing (not shown). Overall, the geometric radius of ice particles is
smaller for convective trajectories than for slow trajectories. Convective trajectories therefore appear to produce clouds with longer lifetimes and greater optical thickness (more and smaller ice particles) than slow trajectories. These clouds also have lower temperatures, since convective trajectories ascend to lower pressures and temperatures (Figure 5 b). Therefore the radiative properties of clouds formed by air parcels in the outflow of the WCB depend on their ascent pathways. The relevance of the different radiative properties depends on the degree to which the WCB outflow clouds are overlaided by in-situ cirrus and
the optical thickness of the latter.

In the hours after the ascent, $RH_i$ decreases faster for slow trajectories than for convective trajectories (Figure 11 d). For both groups the 90 th percentile remains above 100% even after 25 h, but the mean and median for convective trajectories is much larger. This is in part because in the hours after the ascent, convective trajectories keep ascending whereas slow trajec-
tories descend approximately 20 hPa after 20 h (not shown). Convective trajectories however also produce more ice particles that remain in the air parcel for longer and they counteract the formation of subsaturation by sublimating in case the relative humidity goes below 100%.

After the ascent, we can summarise the behavior of trajectories by noting that convective trajectories remain supersaturated
for longer periods (up to more than a day) and retain higher ice number and mass concentrations much longer than slow trajectories.

## 6   Conclusions and Discussion

In this paper, we investigate the transport of moisture into the UTLS by a WCB which occurred on 23 September 2017. The analysis is based on numerical simulations with the ICON model in double nested convective-permitting set-up and makes use
of high-resolution air mass trajectory data from the ICON online trajectory module. Trajectories ascending 600 hPa or more are considered to constitute the WCB.

First, we investigate the ascent behaviour of WCB trajectories. Various ascent-time metrics suggest an abundance of fast, convective ascent in the WCB: 78% of all trajectories experience a 0.5 h period in which the average ascent velocity is larger
than 120 hPa/h. The fast ascent for most trajectories does not extend over the full 600 hPa ascent consistent with recent observations of embedded convection in WCB (Blanchard et al., 2020). A small fraction of WCB trajectories ($\sim 5\%$) complete the entire WCB ascent in few hours. The distribution of 600 hPa ($\tau_{600}$) is similar in structure to earlier investigations of Rasp et al. (2016); Oertel et al. (2021, 2023), although unsurprisingly case-to-case variations in the fraction of very fast ascending trajectories emerge. Hence, (embedded) convection is confirmed as an ubiquitous feature also in the WCB investigated here.






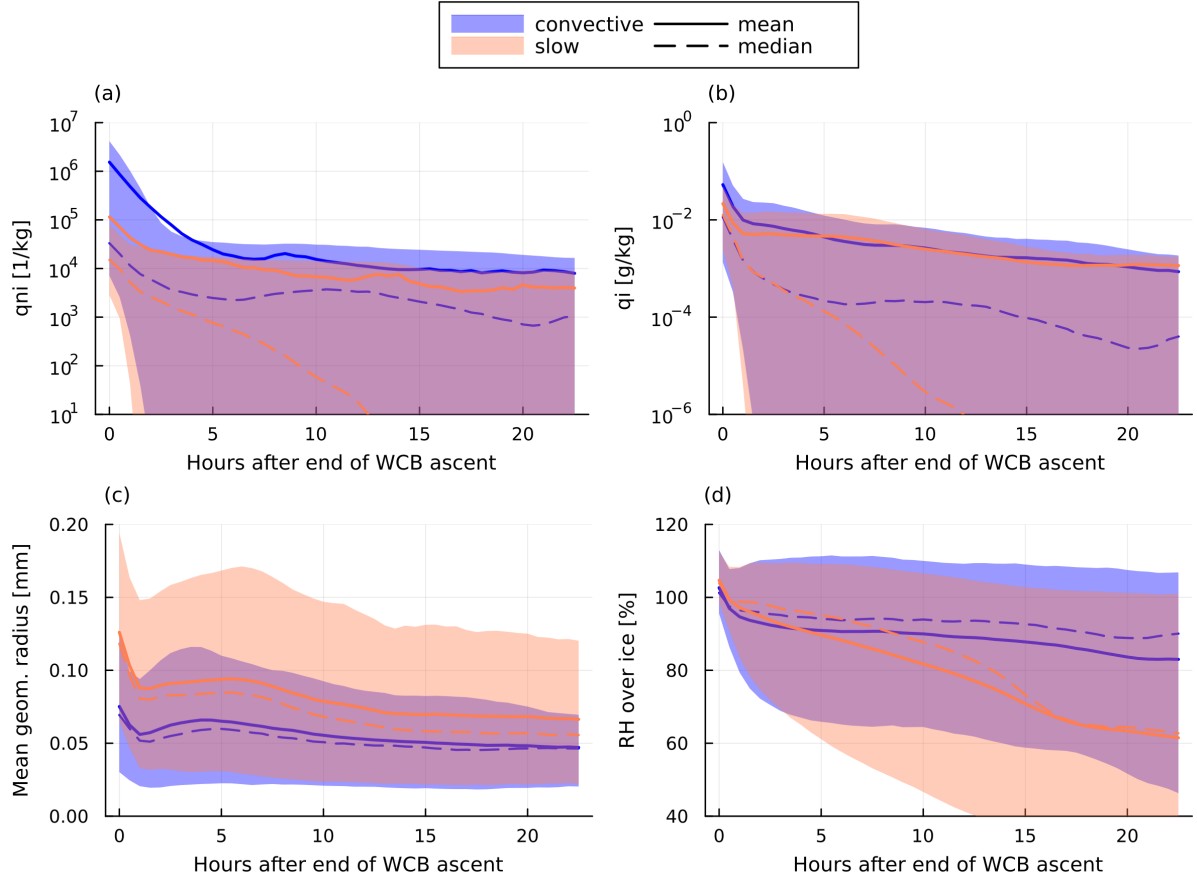

**Figure 11.** The ice number concentration (a), ice mass concentration (b), mean geometric radius of ice particles (c) and relative humidity over ice (d) in the hours after the end of WCB ascent. The mean (solid) and median (dashed) and the 10th and 90th percentiles (shaded) are plotted in blue for convective and orange for slow trajectories.

Next, we analysed how the moisture content and thermodynamic conditions change along the WCB ascent (at start and end of ascent) and consider variations across different ascent timescales. To highlight the impact of different ascent timescale we contrast properties from trajectories with ascent timescales of less than 6 h ("convective" trajectories) and those with ascent timescales more than 20 h ("slow" trajectories). The convective trajectories begin their ascent at higher temperatures and pres-

sures than slow trajectories and therefore have larger specific humidity values at the start of ascent. Only small variations in the initial hydrometeor content are found. By the end of ascent all WCB trajectories have lost more than 95 % of their initial moisture. Nonetheless there is substantial variability in the outflow moisture content: Total moisture vary by roughly a factor ten across all trajectories ($\sim 0.05\,\mathrm{g\,kg^{-1}}$ to $\sim 0.6\,\mathrm{g\,kg^{-1}}$) and about a factor three if the median values per ascent timescale are considered ($\sim 0.1\,\mathrm{g\,kg^{-1}}$ to $\sim 0.3\,\mathrm{g\,kg^{-1}}$). Quantification of moisture loss pathways suggest that the 80 % to 90 % of the

moisture is lost by precipitation formation (and there predominantly through rain), while turbulent mixing results in a moisture




loss of up to 20 %. The turbulent contribution arises mainly at pressures larger than 700 hPa and is more pronounced for slow trajectories.

The variability in end-of-ascent moisture content is mainly due to varying specific humidity, with slow trajectories having larger values than their convective counterparts in contrast to the initial conditions. The primary reason are colder outflow temperatures for convective trajectories and a strong correlation of specific humidity with temperature. Deviations from the thermodynamic equilibrium specific humidity are found to be of second order importance, but maybe of relevance for the further evolution of the outflow cirrus. The fraction of supersaturated trajectories is larger in slow (80 %) than in convective trajectories (56.5 %). It is shown that the final partitioning between water vapor and condensate is strongly influenced by the

temperature at which air parcels fully glaciate and the number of ice crystals they contain at temperatures below about $-20\,^{\circ}C$: Slow trajectories glaciate earlier and contain less ice crystals.

Hydrometeor content at end-of-ascent is generally small. The median hydrometeor content slightly decreases with decreasing ascent timescale, but strongly increases for trajectories with ascent timescales less than about 3 h (the mean increases

for ascent timescales less than about 10 h). The evolution of microphysical properties and precipitation loss during the WCB ascent suggest that the increasing efficiency of condensate removal towards shorter timescales is driven by a transition of the precipitation formation pathway from deposition-aggregation-dominated to riming-dominated. The strong decrease in removal efficiency for the shortest ascent timescales may be driven by a decreasing ratio of ascent timescale to precipitation formation timescale. However, the used output frequency of trajectory data is too small to allow for an in-depth analysis of the micro-

physical evolution of these trajectories.

Finally, we investigate the evolution of WCB outflow after the end of ascent. While hydrometeors contribute little to the total moisture transport to the UTLS (except for ascent timescale less than 10 h), they contribute to the WCB associated cirrus and therefore are important for cloud radiative heating. Convective trajectories contain larger ice crystal number concentra-

tion (median difference about factor two) but similar ice crystal mass than slow trajectories at end-of-ascent. The resulting difference in mean geometric radius impacts the lifetime along with differences in the outflow position and associated vertical motion: The median ice mass mixing ratio in convective outflow cirrus remains larger than 0.1 mg kg for about 15 h, while outflow cirrus derived from slow trajectories median ice mass mixing ratio drops below this value after about 5 h.

The physical processes governing the moisture transport and import into the UTLS are captured by non-dimensional metrics of the Lagrangian moisture budget and provide a concise overview thereof:

– $DR_{mix}$ generally increases with $\tau_{600}$ highlighting stronger moisture loss through turbulent processes



- CR increases with decreasing $\tau_{600}$ reflecting the larger fraction of initial humidity converted to condensate for fast ascending trajectories. This tendency is mainly due to larger vertical displacement and colder outflow temperatures with a secondary contribution of a smaller fraction of supersaturated trajectories for fast ascending trajectories.

- PE slightly increases with decreasing $\tau_{600}$ due to a shift in the dominant precipitation formation pathway towards riming-dominated. For very short $\tau_{600}$ PE decreases substantially resulting in large hydrometeor mass mixing ratio in the WCB outflow.

The findings presented in this paper are broadly consistent with the limited number of earlier studies on the cloud and up-draft structure of WCB clouds (Rasp et al., 2016; Oertel et al., 2019; Binder et al., 2020; Blanchard et al., 2020; Oertel et al., 2021; Blanchard et al., 2021; Oertel et al., 2023). Most of these studies focus on the ascent characteristics and mid-level latent heating distribution. Both modelling and observational case studies suggest significant variability in the vertical velocities with ubiquitous rapid ascent segments. The modelling studies of Oertel et al. (2019, 2021, 2023) suggest different latent heating structure and microphysical pathways for precipitation formation dependent on the local vertical velocity. On a climatological scale, the analysis of CloudSat and CALIPSO data by Binder et al. (2020) supports the ubiquitous presence of rapid ascent segments with different microphysical characteristics, i.e. reflectivity structure. However, we are not aware of an earlier study quantifying the impact of the meso-scale variability in ascent characteristics on the moisture import into the UTLS and WCB outflow cirrus. While the consistency in mid-level cloud characteristics with earlier studies suggests model results presented here are physically plausible, the ICON model and in particular the representation of microphysical processes therein has some important limitations: First, the ICON model uses a saturation adjustment scheme, which enforces water saturated conditions in mixed- and warm-phase cloud regions. Secondly, the parameterisation of ice formation (primary and secondary) still involves large uncertainties, which may influence the glaciation of WCB clouds. Therefore, the verification of our key results with observational data is strongly warranted and is planned for a follow-up study. However, an obstacle to this is the sparsity of high-quality (in-cloud) humidity data in the altitude range between 300 hPa to 100 hPa.

Despite the pending verification with observational data, we think our findings are important as they imply that the use of lower-resolution models with parameterised convection likely not represents the meso-scale variability in moisture transport and cirrus properties. Specifically, they may underestimate the intensity of convection in WCBs, and therefore incorrectly model the transport of hydrometeors and moisture to the UTLS. This could have implications for studies that aim to determine the role of WCBs in the Earth's climate. For instance, Joos, 2019 conducted a WCB climatology using the ERA-Interim dataset (which parameterises convection) and found that WCBs usually have a cooling effect when located further south in the early stages of their development, and a heating effect further north later on. The balance between the cooling and heating effect over the cyclone lifecycle may be affected by the preferential occurrence of convective trajectories in the southern part of the WCB outflow and the associated long-lived cirrus with comparable small effective radii. Additionally, the possible mis- or under-representation of microphysical processes during the WCB ascent could impact studies that investigate the role of WCBs in the formation of precipitation extremes (Catto et al., 2015) and forecast inaccuracies (Grams et al., 2011; Grams



and Archambault, 2016; Martínez-Alvarado et al., 2015). The latter point is supported by the fact that forecast errors generally grow more rapidly in regions with high convective activity (Rasp et al., 2016). This study demonstrates that the meso-scale updraft structure projects onto the moisture structure of the WCB outflow in the UTLS and thereby suggests a pathway for error growth and climatological influence of the meso-scale structure beyond potential vorticity modification. An adequate representation of meso-scale variability seems therefore crucial for a quantification of the impact of WCBs on extratropical weather and climate.

*Code and data availability.* The ICON source code is distributed under an institutional license issued by the German Weather Service (DWD). For more information see https://code.mpimet.mpg.de/projects/iconpublic (DWD and MPI, 2015). The model output of the ICON simulation is available from the authors upon request. All other data will be published upon publication, but can be sent to the reviewers upon request also. The code for the analysis of data and for creating the plots is already accessible through the public gitlab RLP repository https://gitlab.rlp.net/coschwen/schwenk2024_wcb.



**Appendix A: Case Study Plots**

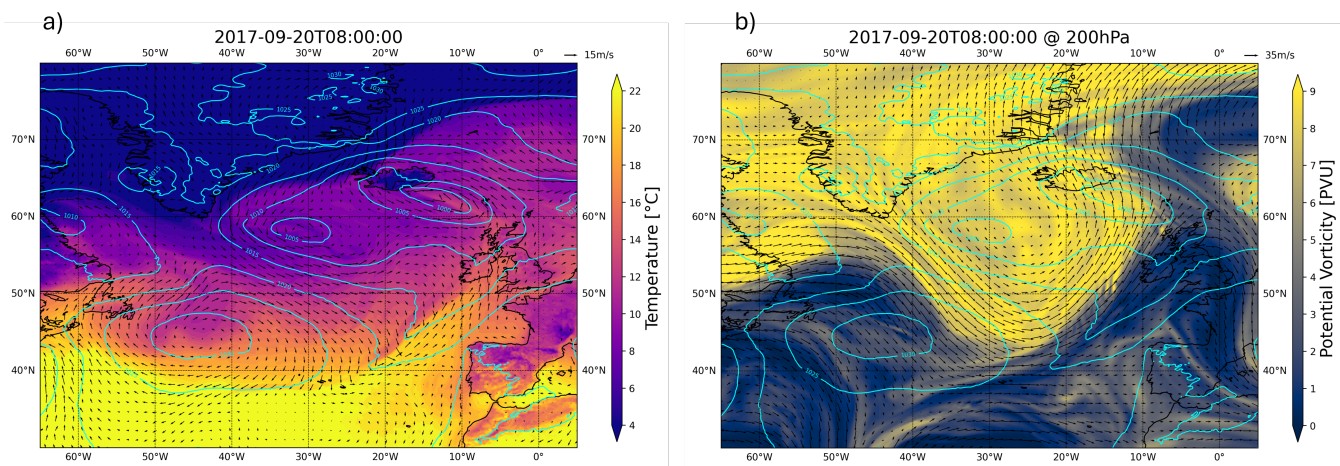

**Figure A1.** Snapshot of the northern Atlantic at 08:00 UTC on 20 September 2017, a) shows the 2 meter temperature (heatmap), sea-level pressure (contours) and 10 meter wind field (arrows); b) shows the upper level PV (heatmap) and wind field (arrows) at 200 hPa and sea-level pressure (contours). The combining winds from the high-pressure system (-45°W, 45°N) and the low-pressure system (-30°W,58°N) blow a cold airmass over the northern Atlantic.

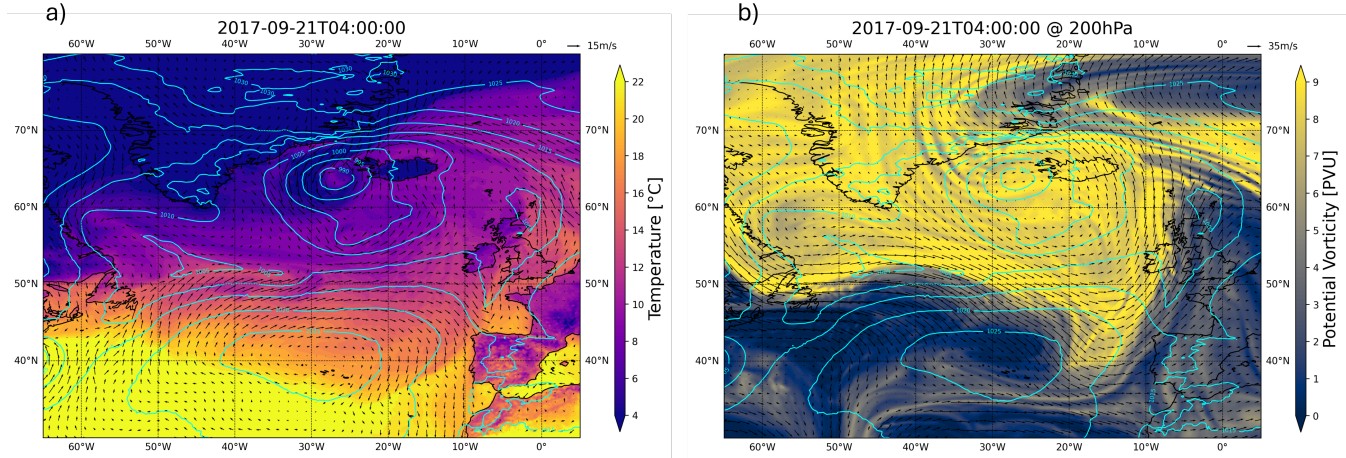

**Figure A2.** Snapshot of the northern Atlantic at 04:00 UTC on 21 September 2017, a) shows the 2 meter temperature (heatmap), sea-level pressure (contours) and 10 meter wind field (arrows); b) shows the upper level PV (heatmap) and wind field (arrows) at 200 hPa and sea-level pressure (contours). The low pressure system starts forming at -45°W and 53°N, where the winds converge. At the same time, an upper level PV ridge (-55°W, 55°N) approaches from the West.



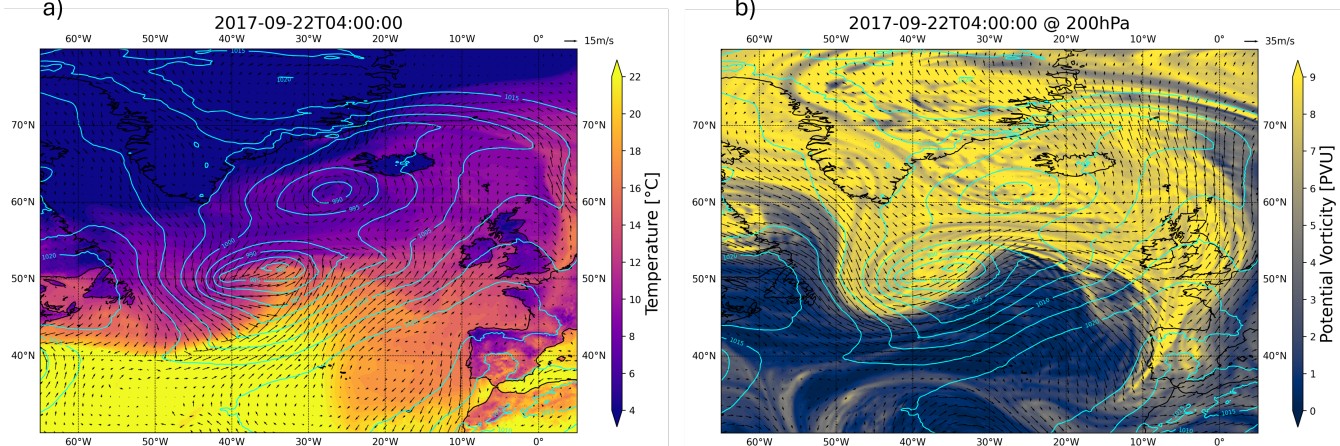

**Figure A3.** Snapshot of the northern Atlantic at 04:00 UTC on 22 September 2017, a) shows the 2 meter temperature (heatmap), sea-level pressure (contours) and 10 meter wind field (arrows); b) shows the upper level PV (heatmap) and wind field (arrows) at 200 hPa and sea-level pressure (contours). The cyclone has undergone explosive cyclogenesis due to the upper level PV anomaly inducing cyclonic rotation in the atmospheric column.

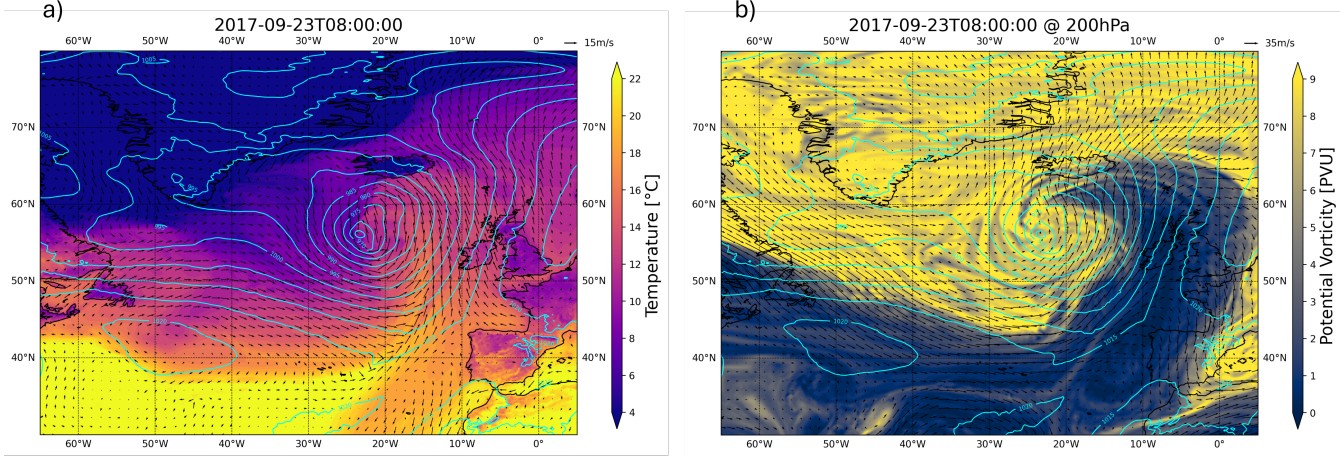

**Figure A4.** Snapshot of the northern Atlantic at 08:00 UTC on 23 September 2017, a) shows the 2 meter temperature (heatmap), sea-level pressure (contours) and 10 meter wind field (arrows); b) shows the upper level PV (heatmap) and wind field (arrows) at 200 hPa and sea-level pressure (contours). The cyclone has assumed the typical WCB comma shape and is transporting low PV into the upper atmosphere. The norther tip of the WCB reaches Iceland.



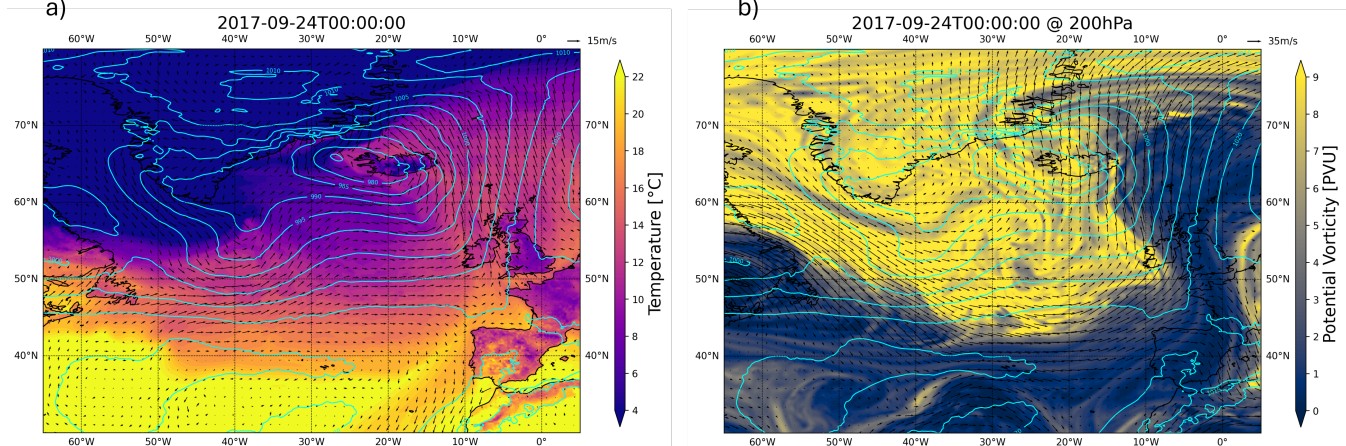

**Figure A5.** Snapshot of the northern Atlantic at 00:00 UTC on 24 September 2017, a) shows the 2 meter temperature (heatmap), sea-level pressure (contours) and 10 meter wind field (arrows); b) shows the upper level PV (heatmap) and wind field (arrows) at 200 hPa and sea-level pressure (contours). The cyclone begins to dissipate over northern Europe and has created an upper level trough.

## Appendix B: Moisture budget terms: definitions and derivations

In this section, we define, describe and/or derive variables that we use in this paper to formulate Lagrangian versions of DR, PE and CR (Section 2.6.2, 2.6.3). Finally, we compare our Lagrangian versions with Eulerian versions from previous studies and show how they can be heuristically related.

### B1 Total moisture content

The total moisture content is defined as follows:

$$Q_{\mathrm{tot}}(t) := \sum_{\mathrm{x}} \mathrm{qx}(t), \qquad \mathrm{x} \in [\mathrm{v,c,r,i,s,g,h}]. \tag{B1}$$

### B2 Moisture removal by mixing processes

The total moisture removed by mixing processes is defined as a function of normalised ascent time $t$ as follows:

$$Q_{\mathrm{tcr}}(t) := Q_{\mathrm{turb}}(t) + Q_{\mathrm{conv}}(t) + \mathfrak{R}(t). \tag{B2}$$

$Q_{\mathrm{turb}}(t)$ is the accumulated moisture removed from the parcel by the turbulent diffusion scheme and is defined as:

$$Q_{\mathrm{turb}}(t) := \sum_{\mathrm{x}} \mathrm{qxturb}(t), \qquad \mathrm{x} \in [\mathrm{v,c,i}]. \tag{B3}$$



$Q_{\mathrm{conv}}(t)$ is the accumulated moisture removed from the parcel by the convection scheme, which is only of consequence when a trajectory is outside either of the nested domains. It is defined as:

$$Q_{\mathrm{conv}}(t) := \sum_{\mathrm{x}} \mathrm{qxconv}(t), \qquad \mathrm{x} \in [\mathrm{v,c,r,i,s}]. \tag{B4}$$

Finally, $\mathfrak{R}(t)$ is the residual term which accounts for the accumulated moisture lost due to numerical and interpolation errors. It is defined as:

$$\mathfrak{R}(t) := Q_{\mathrm{tot}}(0) - Q_{\mathrm{tot}}(t) + Q_{\mathrm{tc}}(t) - P(t). \tag{B5}$$

This term is necessary, because although the online trajectory scheme provides highly accurate data computed at each time step using the Eulerian wind fields from ICON, numerical errors cannot be completely avoided. In each time step, the trajectory equation ($\frac{D\boldsymbol{x}}{Dt} = \boldsymbol{u}(\boldsymbol{x},t)$) is solved to calculate the new trajectory position (Miltenberger et al., 2013). The wind field implied by this solution may differ slightly from that given by ICON, leading to the first possible source of uncertainty. Variables from the ICON Eulerian grid are then linearly interpolated to the new trajectory position, providing the second source of uncertainty. One of the consequences of this is that the moisture budget, calculated by summing the total water content and all possible moisture removal/addition mechanisms, is not closed:

$$Q_{\mathrm{budget}}(t) := \underbrace{\sum_{\mathrm{x=v,c,r,i,s,g,h}} \mathrm{qx}(t)}_{=:Q_{\mathrm{tot}}(t)} - \underbrace{\sum_{\mathrm{x=v,c,i}} \mathrm{qxturb}(t) - \sum_{\mathrm{x=v,c,r,i,s}} \mathrm{qxconv}(t)}_{=:Q_{\mathrm{tc}}(t)} + P(t) \neq \mathrm{const.} \tag{B6}$$

The numerical residual term $\mathfrak{R}(t)$ is defined such that it closes the moisture budget. We can split up the residual into the water vapour and the hydrometeor residual ($\mathfrak{R}_{\mathrm{v}}(t)$ and $\mathfrak{R}_{\mathrm{hy}}(t)$, respectively):

$$\mathfrak{R}(t) = \mathfrak{R}_{\mathrm{v}}(t) + \mathfrak{R}_{\mathrm{hy}}(t). \tag{B7}$$

These terms are defined as follows:

$$\mathfrak{R}_{\mathrm{v}}(t) := Q_{\mathrm{v}}(0) - Q_{\mathrm{v}}(t) - \Delta\mathcal{H}(t) + Q_{\mathrm{v,tc}}(t), \tag{B8}$$

and

$$\mathfrak{R}_{\mathrm{hy}}(t) := Q_{\mathrm{hy}}(0) - Q_{\mathrm{hy}}(t) + \Delta\mathcal{H}(t) + Q_{\mathrm{hy,tc}}(t) - P(t), \tag{B9}$$

where we use $Q_{\mathrm{v,tc}}(t)$, $Q_{\mathrm{hy,tc}}(t)$ and $P(t)$ from Section 2.6 and define $\Delta\mathcal{H}(t)$ as the sum over all microphysical processes that change the vapor/hydrometeor content (written now without time dependence):

$$\Delta\mathcal{H} := \mathrm{qihh} + \mathrm{qcnuc} + \mathrm{cond} + \mathrm{evap} + \mathrm{qxdep} + \mathrm{satad\_II} + \mathrm{revap} + \mathrm{fevap}. \tag{B10}$$

These processes are described in the supporting information.





## B3    Lagrangian net precipitation rate

The Lagrangian net precipitation rate is defined as follows:

$$P(t) := \sum_{\text{x}} \big[\text{qxin}(t) - \text{qxout}(t)\big], \qquad \text{x} \in [\text{r}, \text{i}, \text{s}, \text{g}, \text{h}]. \tag{B11}$$

The variables $\text{qxin}(t)$ and $\text{qxout}(t)$, in the Lagrangian framework of online trajectories, are the time-integrated rates of hydrometeors entering the parcel from above (in) and leaving it at its lower edge (out) at the ascent time $t$. Note that this means that we are not talking about surface precipitation, but precipitation out of the Lagrangian air parcel. We refer to $\text{qxin/out}(t)$ as *fluxes*. Calculating precipitation in this way ensures that only precipitation formed (or grown) *within* the parcel is taken into account. A net precipitation rate of zero therefore means that any precipitation entering the parcel from above does not collect (and thus remove) any additional water, and is equal to the precipitation leaving the parcel at its lower edge.

## B4    Additional microphysical variables for derivation of PE and CR

**Lagrangian hydrometeor growth term** $(C_{\text{hy}})$

The hydrometeor growth term is the sum of all microphysical processes that convert moisture into hydrometeors. It is given by:

$$C_{\text{hy}}(t) = \text{cond}(t) + \text{qcnuc}(t) + \text{qihh}(t) + \text{depo}(t), \tag{B12}$$

where we have the sum of condensation (cond), CCN nucleation (qcnuc), homogeneous and heterogeneous ice nucleation (qihh) and deposition (depo) (see supporting information).

**Lagrangian vapour growth term** $(E_{\text{v}})$

The Lagrangian vapour growth term is the sum over all microphysical processes that increase the vapour content by evaporating ice or sublimating water. It is given by:

$$E_{\text{v}}(t) = \text{evap}(t) + \text{revap}(t) + \text{fevap}(t) + \text{subl}(t), \tag{B13}$$

where we have the evaporation of cloud drops (evap), evaporation of rain (revap), sublimation of snow, graupel and hail (fevap) and the sublimation of ice (subl) (see supporting information).

**Hydrometeor/Water vapour tendency** $(Q_{\text{hy,tc}}(t)/Q_{\text{v,tc}}(t))$

The hydrometeor/water vapour tendency gives the accumulated hydrometeor/water vapour mass transported **out** of the air parcel during the ascent through either the turbulence or the convection scheme. For hydrometeors and vapor, respectively, it is given by

$$Q_{\text{hy,tc}}(t) = \text{qcturc}(t) + \sum_{\text{x}=\text{c,r,i,s}} \text{qxconv}(t), \quad Q_{\text{v,tc}}(t) = \text{qvturc}(t) + \text{qvconv}(t). \tag{B14}$$



Here we use the *corrected turbulence tendencies* qcturc and qvturc that account for the strong instantaneous compensation between process rates from the turbulence scheme and the second call to the saturation adjustment in ICON (see supporting information).

**Net initial hydrometeor/vapor content** (HYD and VAP)

Using these variables (and the numerical hydrometeor/water vapour residuals $\mathfrak{R}_{\mathrm{hy}}(t)$ and $\mathfrak{R}_{\mathrm{v}}(t)$), we formulate the net initial hydrometeor content:

$$\mathrm{HYD}(t) := Q_{\mathrm{hy}}(0) - Q_{\mathrm{hy,tc}}(t) - \mathfrak{R}_{\mathrm{hy}}(t) \qquad \left(\text{with} \quad Q_{\mathrm{hy}}(t) := \sum_{\mathrm{x=c,r,i,s,g,h}} \mathrm{qx}(t)\right), \tag{B15}$$

and the net initial vapour content:

$$\mathrm{VAP}(t) := Q_{\mathrm{v}}(0) - Q_{\mathrm{v,tc}}(t) - \mathfrak{R}_{\mathrm{v}}(t) \qquad \left(\text{note}: \quad \mathrm{HYD}(t) + \mathrm{VAP}(t) = Q_{\mathrm{tot}}(0) - Q_{\mathrm{tcr}}(t)\right). \tag{B16}$$

These terms are almost equal to the initial hydrometeor/vapor content, but take into account additional hydrometeors/vapor transported in or out of the parcels by the turbulence or convection scheme (or numerical residuals). We create these terms because we follow the following logic: hydrometeors that precipitate out of a parcel but were brought in by turbulence, for
example, should not be counted as precipitation formed within the parcel. Using only the initial hydrometeor/vapor contents would mean that we could, theoretically, see more precipitation "formation" than we have initial total water, which would be confusing.

### B5 Comparison of PE, CR and DR to Eulerian variables

We can compare our Lagrangian definitions of PE, CR and $\mathrm{DR}_{\mathrm{mphys}}$ to the Eulerian definitions from Miltenberger (2014) by
835 making simplifications. If we let $Q_{\mathrm{tcr}}(1)$ and $Q_{\mathrm{hy}}(0)$ go to zero (i.e, we neglect the initial hydrometeor content, all numerical residuals and all transport by the turbulence and convection scheme), we get:

$$\lim_{Q_{\mathrm{tcr}}(1),Q_{\mathrm{hy}}(0)\to 0} \mathrm{DR}_{\mathrm{mphys}} = \frac{P(1)}{C_{\mathrm{hy}}(1) + E_{\mathrm{v}}(1) + \cancel{\mathrm{HYD}(1)}^{\,0}} \cdot \frac{C_{\mathrm{hy}}(1) + E_{\mathrm{v}}(1) + \cancel{\mathrm{HYD}(1)}^{\,0}}{\cancel{\mathrm{VAP}(1)}} \cdot \frac{\cancel{\mathrm{VAP}(1)}}{Q_{\mathrm{tot}}(0) - \cancel{Q_{\mathrm{tcr}}(1)}^{\,0}} \tag{B17}$$

which reduces to:

$$\mathrm{DR}_{\mathrm{mphys}}^{\mathrm{Eul}} = \underbrace{\frac{P(1)}{C_{\mathrm{hy}}(1) + E_{\mathrm{v}}(1)}}_{\mathrm{PE_{Eul}}} \cdot \underbrace{\frac{C_{\mathrm{hy}}(1) + E_{\mathrm{v}}(1)}{Q_{\mathrm{tot}}(0)}}_{\mathrm{CR_{Eul}}} = \mathrm{PE_{Eul}} \cdot \mathrm{CR_{Eul}}. \tag{B18}$$

These definitions are similar to those from Miltenberger (2014). The simplifications we have made also result in $\mathrm{DR}_{\mathrm{mix}} = 0$, meaning that $\mathrm{DR} = \mathrm{DR}_{\mathrm{mphys}}^{\mathrm{Eul}}$. We have thus shown that our Lagrangian definitions of DR, PE and CR can be understood as generalizations of those in the Eulerian framework (or alternatively, the Eulerian definitions from Miltenberger (2014) are special cases that neglect initial hydrometeor contents, numerical residuals and transport by turbulence and convection schemes).





## Appendix C: General plots

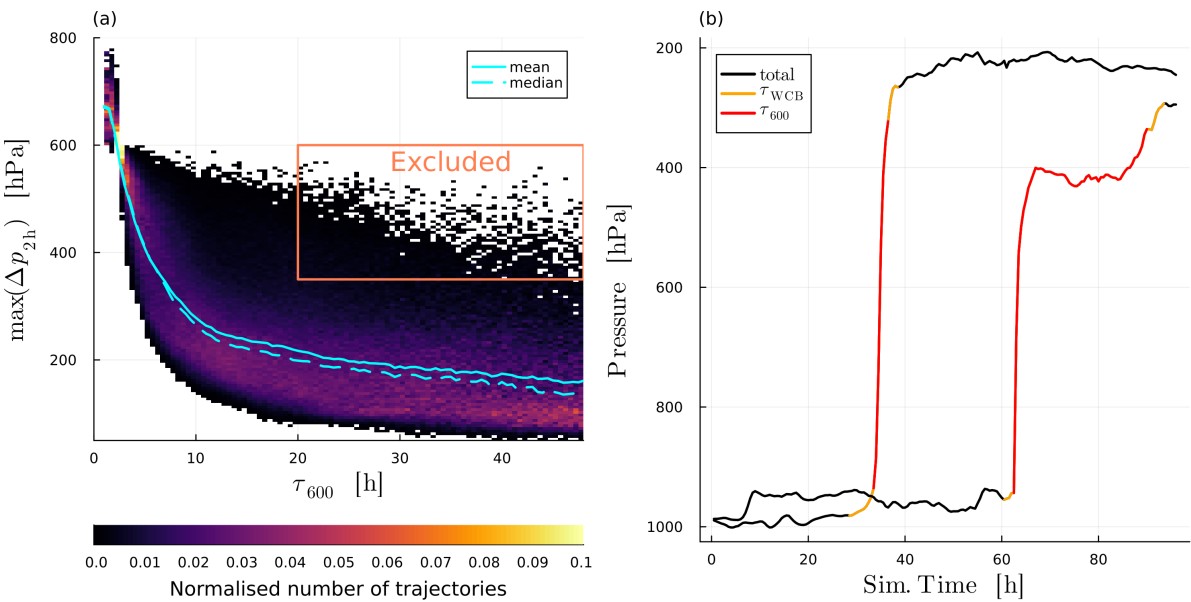

**Figure C1.** a) the pressure course for two example trajectories, one convective and one slow, showing that some trajectories ascend convectively but not for the entire 600 hPa. b) the maximum pressure velocity over two hours for each trajectory ($\max(\Delta p_{2\,\mathrm{h}})$) as a 2-dimensional normalised trajectory over $\tau_{600}$. Trajectories that fall into the orange square are excluded from the analysis.

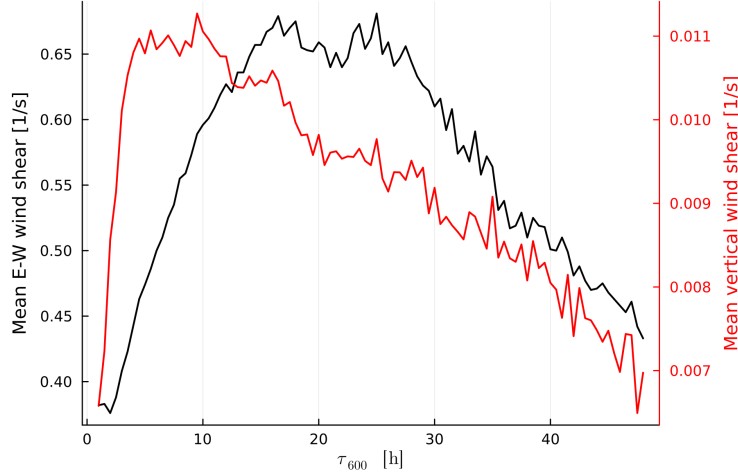

**Figure C2.** Mean horizontal (black, left y-axis) and vertical (red, right y-axis) wind shear for all data points along WCB ascent for trajectories over $\tau_{600}$.




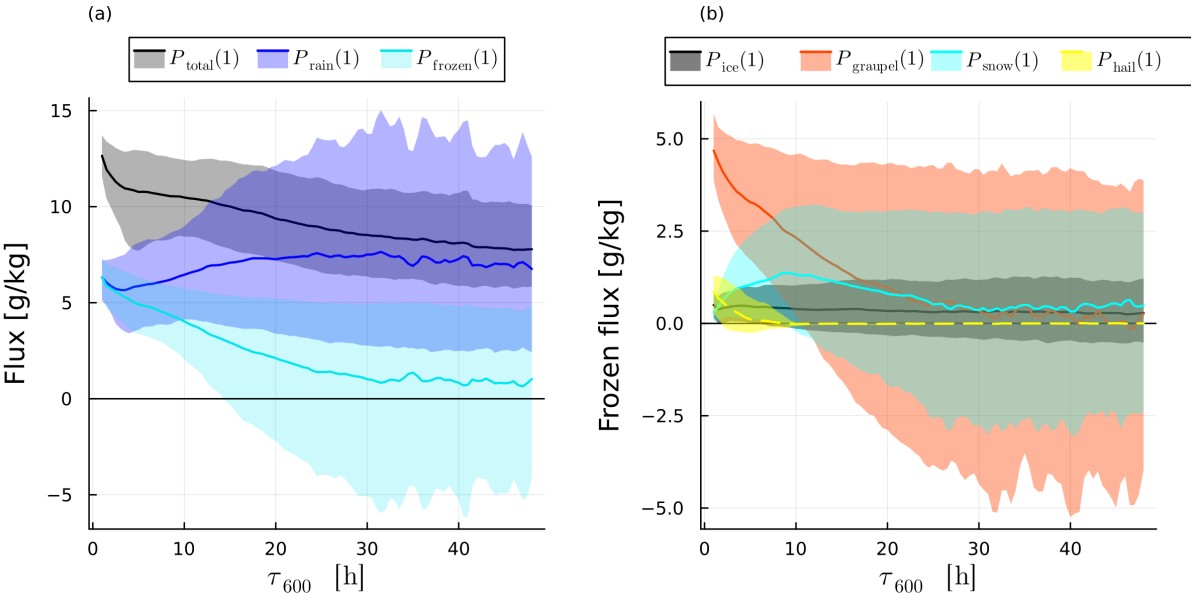

**Figure C3.** In (a) the median total (black), warm-phase (blue) and cold-phase (cyan) precipitation flux by the end of the ascent is shown, with 10 th and 90 th percentiles shaded. In (b) the individual contributions to the cold-phase flux are shown. Ice (black), graupel (red), snow (cyan) and hail (yellow dashed, note: zero everywhere except for $\tau_{600} < 10$ h.)

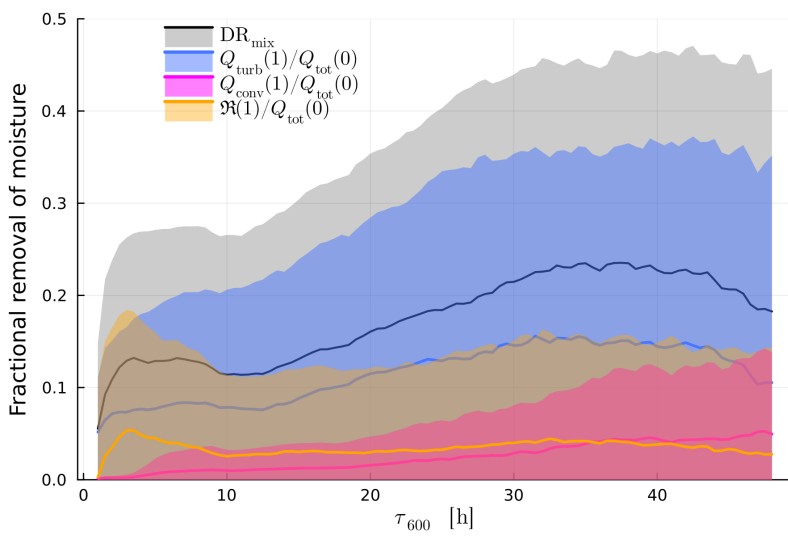

**Figure C4.** Mean $DR_{mix}$ (black) with individual contributions from turbulence (blue), convection (red) and numerical residual (yellow). 10 th and 90 th percentiles shaded.



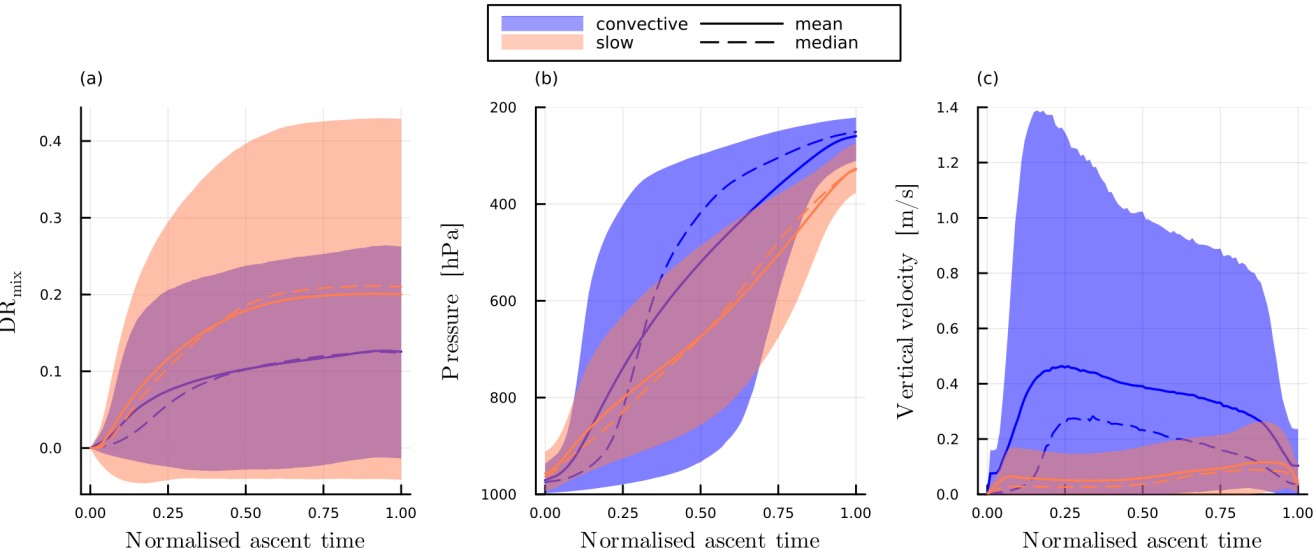

**Figure C5.** $DR_{mix}$ (a), pressure (b), and vertical velocity (c) over normalised ascent time. In all panels the mean (solid) and median (dashed) as well as the 10th and 90th percentiles (shaded) are plotted in blue for convective and orange for slow trajectories.







**Figure C6.** Time integrated evaporation (a), Wegener-Bergeron-Findeisen (b), deposition (c) and riming rate (d) over normalised ascent time. In all panels the mean (solid) and median (dashed) as well as the 10th and 90th percentiles (shaded) are plotted in blue for convective and orange for slow trajectories.



*Author contributions.* CS and AM designed the experiment and conducted the numerical simulations. AM implemented the online diagnostics, CS wrote the post-processing code. CS and AM worked jointly on the interpretation of the results. CS drafted the final manuscript with contributions from AM.

*Competing interests.* The authors have no competing interests.

*Acknowledgements.* This work was funded by the Deutsche Forschungsgemeinschaft (DFG, German Research Foundation) – TRR 301
– Project-ID 428312742: "The tropopause region in a changing atmosphere", sub-project B08 coordinated by Annette Miltenberger. The
authors gratefully acknowledge the computing time granted on the supercomputer MOGON 2 at Johannes Gutenberg University Mainz
(hpc.uni-mainz.de), which is a member of the AHRP (Alliance for High Performance Computing in Rhineland Palatinate, www.ahrp.info)
and the Gauss Alliance e.V. We further thank Annika Oertel for useful discussion input and sharing her ICON set-up used in Oertel et al.
(2023).



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
