# Peer review of "The role of ascent timescale for WCB moisture transport into the UTLS"

_EGUsphere, 2024_

## Author Response (AR1)

**Manuscript Revisions Summary**

Below is a list that summarises our responses to the RCs as well as the most important changes we have made in the manuscript. The detailed responses are appended at the end.

**1. Readability and Redundancy Reduction (RC1 and RC2)**

Sections 5.2, 5.3, 5.4, and 5.5: These sections have been shortened, with key points restructured to improve readability and reduce redundancy. Essential findings were retained to ensure a comprehensive understanding of the study.

**2. Clarification on Ascent Timing (RC1)**

Lines 182-189: Clarified that t=0 marks the beginning of the tau_WCB ascent, defined as the period around tau_600 where ascent velocity is at least 8 hPa/h. This criterion ensures a consistent focus on significant ascent, avoiding periods of minimal vertical motion.

**3. Clarification of Figure References (RC1)**

Lines 369-370: Added clarification on the result discussed, indicating it is presented in Figures 4 and C1a, enhancing readability and providing a clearer reference for the reader.

**4. Hypothesis on Buoyant Acceleration (RC1)**

Lines 387-389: Added a citation to Schäfler and Harnisch (2014) supporting the proposed hypothesis on buoyant acceleration, without performing explicit calculations for buoyant acceleration. Minor amendments clarify this point.

**5. Ice Supersaturation and Key Takeaway Points (RC1)**

Lines 414-428: Revised and structured this section into sub-paragraphs for improved readability. Emphasized the key points: (i) thermodynamic conditions at the end of ascent primarily determine vapor content, and (ii) RHi distributions differ between convective and slow trajectories due to differences in ice content.

**6. Turbulent Mixing and Planetary Boundary Layer Influence (RC1)**

Lines 470-472: Added a clarification on the impact of time spent in the planetary boundary layer on moisture loss through turbulent mixing. Also clarified for RC1 what we intended to convey.

**7. Precipitation Efficiency (PE) and Temperature Correlation (RC1)**

Line 524: Explained that while convective parcels may have lower temperatures, there is no significant correlation between PE and temperature. Clarified that lower temperatures do not directly imply a higher fraction of hydrometeors will precipitate by the end of ascent, hence not worth mentioning further.

**8. Additional Citation (RC1)**

Lines 611-612: Provided a citation for the statement regarding a finding in the text, addressing an oversight.

**9. Conclusion Section Revision (RC1)**

Lines 700-708: Responded to reviewer feedback on potentially less relevant results, emphasizing the specific relevance for cloud microphysics research. Offered to clarify that this section holds particular interest for readers focused on microphysics.

**10. Applicability of Findings to Broader WCB Population (RC1)**

Lines 726 and onward: Added a discussion on the representativeness of the case study, acknowledging that while this study focuses on a typical, non-extreme open-ocean WCB, further research is required to determine the applicability to a broader WCB population. Mentioned that ongoing model sensitivity experiments and comparisons with observational data will address the impact of specific parameterisations on key findings.

**11. Technical Corrections (RC1)**

Figures A1-A5: Adjusted color schemes and other visual parameters to enhance accessibility and clarity for color-deficient readers, based on reviewer feedback on Figures A*.

**12. Clarification on Chy and VAP terms(RC2)**

Lines 257-256: added clarification on specific terms and where more detailed definitions can be found.

**13. Changed colors in Figure 6 (RC2)**

Figure 6: Changed colors in Figure 6 to enhance visibility
* * *
**Detailed response to RC1:**

Thank you for your thoughtful and positive feedback on our manuscript. We are pleased to hear that you found the analysis interesting and the results novel, especially in relation to convective trajectories and their microphysical properties. Below, we address your specific comments in detail

**There were many sections that I had to read multiple times to ensure that I was understanding the main points, thus it might be worthwhile to consider trimming back some of these calculations and results to the main points that you want the audience to take away**

Thank you for this valuable feedback. This is a point that both reviewers have made, therefore we have carefully reviewed the manuscript with a focus on improving readability and reducing redundancy. We have shortened sections 5.2, 5.3, 5.4 and 5.5 and restructured

key points to enhance clarity. However, we have retained certain sections that may seem extensive, as they contain what we believe are essential findings necessary for a comprehensive understanding of this study. We hope these revisions strike a balance between conciseness and thoroughness.

**Lines 182-189: The paper would benefit from a discussion of how the ascent time begins and ends. It appears this is when w=0, but that is not clear. If this is true, Is there any issue in looking for the time when w=0? It seems like the parcel could undergo very small vertical motion for a long time here. This seems to be supported by some calculations here. Should that impact the normalized time coordinate?**

Thank you for your observation. We have clarified that t=0 specifically marks the beginning of the tau_WCB ascent (see line 187-188 in the revised manuscript), which, as defined earlier in the paper, is the period surrounding tau_600 during which the ascent velocity is at least 8 hPa/h . This criterion ensures that we capture the main ascent in a consistent fashion across parcels, but avoid parcels experiencing very small vertical motion over extended periods. We took great care in the data analysis to ensure that the selection of tau_WCB properly captures significant ascent, avoiding the inclusion of periods with minimal vertical movement.

**Lines 369-370: Is this result shown in a figure?**

Yes, this result is discussed in the preceding paragraphs, which reference Figures 4 and C1a .We believe this result is clearly presented in the figures, and that the current format provides sufficient readability and clarity.

**Lines 387-389: This statement is reasonable, but have the authors performed any calculations to confirm this hypothesis? It seems like you could compute a buoyant acceleration for the trajectory.**

Thank you for your suggestion. While we did not perform explicit calculations for buoyant acceleration, we proposed this explanation based on well-established physical principles. If the reviewer deems it necessary, we could calculate a 'potential buoyancy' for each trajectory, making reasonable assumptions about latent heat release and environmental conditions during ascent. However, we believe that this would add unnecessary complexity for a point that aligns with widely accepted understanding. We have however added a citation to a study that makes a similar finding: see line 391, (Schäfler and Harnisch, 2014) in the revised manuscript.

**Lines 414-428: I am not sure that I follow this discussion about ice supersaturation and what is the takeaway points. Please revise.**

Thank you for your feedback. We have revised the paragraph (now in lines 415-435) and have introduced sub-paragraphs to improve readability and clarify the key takeaway points. Specifically, we aim to highlight: i) that the thermodynamic conditions at the end of the ascent (temperature, pressure) are the primary factors determining vapor content, and ii) that RHi distributions differ between convective and slow trajectories, with higher RHi in slow trajectories likely due to differences in ice content at the end of the ascent. We hope these changes make the discussion more comprehensible and the main points clearer.

**Lines 470-472: This result would seem to suggest that the amount of turbulent mixing is a function of how long the parcel spends in the planetary boundary layer. One might expect a slower trajectory to spend more time there compared to a fast-ascending trajectory. Have you investigated this point?**

Thank you for the insightful comment. Indeed, the amount of moisture lost to turbulence increases with the time spent in the planetary boundary layer, which explains why slower trajectories lose more moisture to turbulent mixing than faster ones. However, the lines in question refer to a peculiar behavior we observed in the *slowest* trajectories, where moisture loss to turbulent mixing decreases again. To investigate this, we calculated the average wind shear experienced by trajectories in each ascent-time bin and found that the slowest ascending trajectories encounter less wind shear (Fig C5). This suggests that the slowest trajectories may be more embedded within the large-scale, coherent flow of the WCB, rather than being influenced by smaller-scale turbulence.

We have made some minor amendments to the paragraph (now in lines 460-470) to provide additional clarification and hope that these revisions make this point more easily understandable.

**Line 524: Is this due to the convective parcels being at a lower temperature? Is this worth noting here?**

We believe the reviewer is asking whether the lower temperatures of convective parcels might explain why precipitation efficiency (PE) decreases with shorter ascent times. Our analysis found no significant correlation between PE and temperature. While lower temperatures mean that the parcel can hold less vapor, this does not necessarily imply that a higher fraction of hydrometeors formed during the ascent must precipitate by the end (which is what PE quantifies). This is therefore not worth mentioning here.

**Lines 611-612: Provide a citation for this statement?**

This was an oversight and we have added the relevant citation (line 598 of the revised manuscript)

**Lines 700-708: For me, these results were potentially less interesting than others. Perhaps remove this from the conclusions section?**

Thank you for your feedback. The lines in question present key findings of the paper which are of particular relevance for those with a primary interest in cloud microphysics. This might only appeal to a subset of readers, but is no less important. If the reviewer feels this distinction is not clear, we can add a clarification to highlight that this part is of specific interest to those focused on cloud microphysics.

**This study is based on a single case study. How representative is this of WCBs as a whole? Is this case an example with similar amounts of convection to a typical WCB? I think it is worth commenting on how broadly these results might be applicable to other WCBs.**

Thank you for raising this important point. In this paper, our focus was not specifically on identifying the most climatologically representative WCB, but rather on analyzing a typical, non-extreme WCB event to explore moisture transport and the differences between convective and non-convective trajectories. We intentionally selected an open-ocean WCB for this study, as the majority of WCBs ascend over the open ocean. While we believe our findings offer valuable insights into WCB dynamics, we acknowledge that further research is needed to determine how broadly these results apply to the wider WCB population and if the results found in this paper can be verified by measurements. We are currently addressing these points for future publications. We have added the following after line 726 in the revised manuscript:

> *The results presented in this paper are strictly only valid for the analysed WCB case, which we have selected to represent a typical, non-extreme event of an open-ocean WCB. While we believe our findings provide valuable insights into WCB moisture transport and may be applicable to similar WCB cases, the extent to which they apply to the larger WCB population remains to be addressed in future studies. Furthermore, some of the findings summarised above may depend on the particular parameterisation set-up used in the analysed simulation, such as the specific microphysics scheme and the absence of a deep convection parameterisation. The influence of these model-specific factors on our key findings is currently being assessed through model sensitivity experiments and comparisons with observational data.*

**Response to Technical Corrections**

Thank you for pointing out these small errors; we have corrected them. Regarding Figures A*, we have adjusted the color scheme and other parameters to improve visual accessibility while ensuring that the plots remain clear for color-deficient viewers.
* * *
**Detailed response to RC2:**

Thank you for your positive feedback and for finding our work suitable for publication in ACP. We are glad that you found the investigation of WCB trajectories and their characteristics interesting. We also thank you for your helpful comments and for identifying the technical errors (points 3-6 and 8-11) which we have now corrected. Below we respond to your comments 1), 2) and 7).

**1) I think this paper is a bit too long and feels redundant. By shortening the main text and organizing the key points, I believe it will become easier for readers to understand.**

Thank you for this valuable feedback. This is a point that both reviewers have made, therefore we have carefully reviewed the manuscript with a focus on improving readability and reducing redundancy. We have shortened sections 5.2, 5.3, 5.4 and 5.5 and restructured key points to enhance clarity. However, we have retained certain sections that may seem extensive, as they contain what we believe are essential findings necessary for a comprehensive understanding of the study. We hope these revisions strike a balance between conciseness and thoroughness.

**2) There is no explanation for the terms of Chy and Ev.**

Thank you for pointing out the potential for confusion. We have added a brief explanation of these terms and adjusted the text to clarify that their precise definitions can be found in Appendix B4.

**7) Figure 6: Because it is hard to find aqua lines in the figure, I recommend change the color of those lines.**

We have replaced the aqua color with light-green to enhance the visibility of the vapor line while ensuring accessibility for color-deficient viewers.